# Improving Aspect Ratio Distribution Fairness in Few-Shot Detector Pretraining via Cooperating RPN's

## Abstract

Region proposal networks (RPN) are a key component of modern object detectors. An RPN identifies image boxes likely to contain objects, and so worth further investigation. An RPN false negative is unrecoverable, so the performance of an object detector can be significantly affected by RPN behavior, particularly in low-data regimes. The RPN for a few shot detector is trained on base classes. Our experiments demonstrate that, if the distribution of box aspect ratios for base classes is different from that for novel classes, errors caused by RPN failure to propose a good box become significant. This is predictable: for example, an RPN trained on base classes that are mostly square will tend to miss short wide boxes. It has not been noticed to date because the (relatively few) standard base/novel class splits on current datasets do not display this effect. But changing the base/novel split highlights the problem. We describe datasets where the distribution shift is severe using PASCAL VOC, COCO, and LVIS datasets.

We show that the effect can be mitigated by training multiple distinct but cooperating specialized RPNs. Each specializes in a different aspect ratio, but cooperation constraints reduce the extent to which the RPNs are tuned. This means that if a box is missed by one RPN, it has a good chance of being picked up by another. Experimental evaluation confirms this approach results in substantial improvements in performance on the ARShift benchmarks, while remaining comparable to SOTA on conventional splits. Our approach applies to any few-shot detector and consistently improves performance of detectors.

## 1 Introduction

Most state-of-the-art object detectors follow a two-stage detection paradigm. A region proposal network (RPN) finds promising locations, and these are passed through a classifier to determine what, if any, object is present. In this architecture, if an RPN makes no proposal around an object, the object will not be detected. For a few-shot detector, one splits the classes into base and novel, then trains the RPN and classifier on base classes, fixes the RPN, and finally fine-tunes the classifier on novel classes using the RPN's predictions.

Objects in large-scale object detection datasets (*e.g.* COCO (Lin et al., 2014); LVIS (Gupta et al., 2019)) have typical aspect ratio that varies somewhat from instance to instance, and often differs sharply from category to category. As a result, the few-shot training procedure has a built-in problem with distribution shift. This phenomenon is illustrated in Figure 1. Imagine all base classes are roughly square, and all novel classes are either short and wide, or tall and narrow. The RPN trained on the base classes should miss some novel class boxes. These boxes will have two effects: the training data the classifier sees will be biased against the correct box shape; and, at run time, the detector may miss objects because of RPN failures. We refer to this problem as the *bias* (the RPN does not deal fairly with different aspect ratios). The bias occurs because the RPN sees few or no examples of the novel classes during training (Kang et al., 2019; Wang et al., 2020; Yan et al., 2019).

To date, this bias has not been remarked on. This is an accident of dataset construction: the standard base/novel splits in standard datasets do not result in a distribution shift. But other base/novel splits do result in a distribution shift large enough to have notable effects, and Section 3 shows our evidence

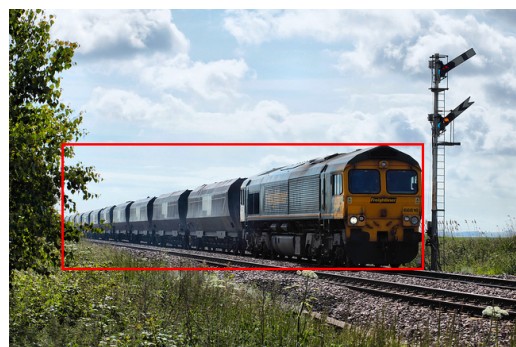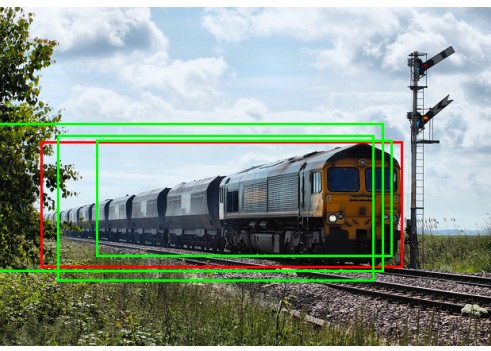

Figure 1: RPN is severely affected by the distribution shift of object aspect ratios from base to novel classes, leading to degenerated few-shot detection performance. After training on the base classes which are mostly *boxy* objects (`bike`, `chair`, `table`, `tv`, `animals`, *etc.*), (**Left**) state-of-the-art few-shot detector DeFRCN (Qiao et al., 2021) built on the conventional RPN misses the *elongated* novel class (`train`) object at the proposal stage and generates no proposal with $\text{IoU}_{\text{gt}} > 0.7$ – this is a disaster (the classifier of DeFRCN will not see a `train` box proposal, and so it cannot detect the `train`). By contrast, (**Right**) our CoRPN's remedy this issue and thus improve few-shot detection for DeFRCN. Red boxes are the groundtruth box of the novel class `train` object; green boxes are box proposals output by the model. We plot positive proposals with $\text{IoU}_{\text{gt}} > 0.7$ following Qiao et al. (2021).

that this effect occurs in practice. In particular, we describe ARShift benchmarks that simulate the real-world scenario where the aspect ratio distribution shift is severe. RPNs in state-of-the-art few-shot detectors are heavily biased towards familiar aspect ratio distributions, and so have weaker than necessary performance on non-standard splits because their RPNs are unfair. Evaluation practice should focus on performance under hard splits.

In few-shot detection applications, a more robust RPN will be more reliable, because applications typically offer no guarantees about the aspect ratio of novel classes. We show how to build a more robust RPN by training multiple RPN classifiers to be *specialized but cooperative*. Our CoRPN's can specialize (and a degree of specialization emerges naturally), but our cooperation constraints discourage individual RPN classifiers from overspecializing and so face generalization problems. CoRPN's are competitive with SOTA on widely used conventional benchmarks of few-shot detection, using conventional splits. But on our ARShift benchmarks with hard splits based on PASCAL VOC, MS-COCO, and LVIS (Everingham et al., 2010; Lin et al., 2014; Kang et al., 2019; Wang et al., 2020), they beat SOTA, because they are more robust to shifts in aspect ratio distribution.

**Our contributions:** (1) We show the bias has severe effects on detector performance, and describe ARShift benchmarks that evaluate these effects. (2) We describe a general approach to improving RPN robustness to distribution shifts. Our CoRPN construction works with many types of few-shot detectors. (3) We show that performance improvements resulting from CoRPN's results from improved fairness. (4) Our CoRPN's are competitive with SOTA on widely used conventional benchmarks. But on the hard splits in ARShift, they beat SOTA, because they are fair.

## 2 RELATED WORK

**Object Detection with Abundant Data.** There are two families of detector architecture, both relying on the fact that one can quite reliably tell whether an image region contains an object independent of category (Endres & Hoiem, 2010; van de Sande et al., 2011). In serial detection, a proposal process (RPN in what follows) offers the classifier a selection of locations likely to contain objects, and the classifier labels them. This family includes R-CNN and its variants (Girshick, 2015; Girshick et al., 2014; He et al., 2017; Ren et al., 2015) In parallel detection, there is no explicit proposal step; these methods can be faster but the accuracy may be lower. This family includes YOLO and its variants (Bochkovskiy et al., 2020; Redmon & Farhadi, 2017; Redmon et al., 2016; Redmon & Farhadi, 2018), SSD (Liu et al., 2016), point-based detectors such as CornerNet (Law & Deng, 2018) and ExtremeNet (Zhou et al., 2019), and emerging transformer-based methods exemplified by DETR (Carion et al., 2020). This paper identifies an issue with the proposal process that can impede strong performance when there is very little training data (the *few-shot* case). The effect is described in the context of two-stage detection, but likely occurs in one-stage detection too.

**Few-Shot Object Detection.** Few-shot detection involves detecting objects for which there are very few training examples (Chen et al., 2018; Kang et al., 2019; Schwartz et al., 2019), and state-of-the-art methods are usually serial (Wang et al., 2019; Yan et al., 2019; Wang et al., 2020; Fan et al., 2020; Wu et al., 2020; Xiao & Marlet, 2020; Yang et al., 2020; Li et al., 2021a; Hu et al., 2021; Zhang et al., 2021; Li et al., 2021b; Zhu et al., 2021). There is a rich few-shot classification literature (roots in Thrun (1998); Fei-Fei et al. (2006)). Dvornik et al. (2019) uses ensemble procedures for few-shot classification. As to detection, TFA (Wang et al., 2020) shows that a simple two-stage fine-tuning approach outperforms other complex methods. Much work seeks improvements by applying different techniques, such as meta-learning (Wang et al., 2019; Yan et al., 2019; Hu et al., 2021; Zhang et al., 2021), metric learning (Han et al., 2021; Wu et al., 2021; Yang et al., 2020), refinement (Wu et al., 2020; Li et al., 2021b), feature reweighting (Kang et al., 2019), semantic relations (Zhu et al., 2021), augmentation (Li & Li, 2021; Zhang & Wang, 2021), and margin loss (Li et al., 2021a). Other work (Fan et al., 2021) alleviates forgetting of base classes. In particular, Qiao et al. (2021) achieves state-of-the-art performance by decoupling the gradient of the backbone and other components of the detector, as well as adding a prototypical calibration module. Here we focus on the two most representative methods – the state-of-the-art DeFRCN (Qiao et al., 2021) and the widely used TFA (Wang et al., 2020) – as our main baselines.

**Few-Shot Detection Benchmarks.** The existing literature can be seen as variations on a standard detection framework, where one splits data into two sets of categories: base classes $C_b$ (which have many training examples) and novel classes $C_n$ (which have few). The RPN and classifier are trained with instances from the base classes, and then fine-tuned with the few-shot novel class data. While the choice of the split can be important in revealing different aspects of few-shot detection, existing benchmarks (Kang et al., 2019; Wang et al., 2020) have only focused on *a few fixed, rather arbitrary* splits. However, we explore the scenario where there exists a notable distribution shift between base and novel classes, and investigate the behavior of RPNs accordingly.

**Proposal Process in Few-Shot Detection.** Relatively little work adjusts the proposal process, which is usually seen as robust to few-shot issues because there are many base examples. (Sun et al., 2021) introduces contrastive-aware object proposal encodings to facilitate classification. Attention mechanisms are also introduced that feed category-aware features instead of plain image features into the proposal process (Hsieh et al., 2019; Fan et al., 2020; Xiao & Marlet, 2020; Osokin et al., 2020), as well as re-ranking proposals based on similarity with query images (Hsieh et al., 2019; Fan et al., 2020). Making the RPN category-aware improves the quality of novel class proposals, *but at inference time the model suffers from catastrophic forgetting of base categories* – current category-aware features cannot summarize the very large number of base class examples efficiently or accurately. An RPN that is generally well-behaved can still create serious trouble in the few-shot case by missing important proposals for the novel classes during fine-tuning. We show that the proposal process can be improved by a carefully constructed cooperating RPN's without substantial loss of performance for the base classes.

## 3 OUR APPROACH

We believe that improving the population of RPN boxes seen by the classifier in training will always tend to improve a detector, and so focus on finding and fixing the effect within a standard few-shot object detection framework. Our proposed strategy is general and can work with different types of few-shot detectors. Here we consider the two most representative methods: the state-of-the-art DeFRCN (Qiao et al., 2021) and the widely used TFA (Wang et al., 2020). We first observe the box aspect ratio distribution shift problem through our pilot study, and show that naïve ensemble of RPN experts does not sufficiently solve this problem on our hard ARShift splits. Then we introduce our CoRPN's that effectively tackles the aspect ratio distribution shift problem via the cooperation and diversity losses.

### 3.1 BACKGROUND

We use the few-shot detection setting introduced in Kang et al. (2019). We split the dataset into two sets of categories: base classes $C_b$ and novel classes $C_n$. The training process is two-phase: (1) base classes training, and (2) fine-tuning with novel classes. In phase 1, the model is trained with base class instances which results in a $|C_b|$-way detector. After base classes training, weights

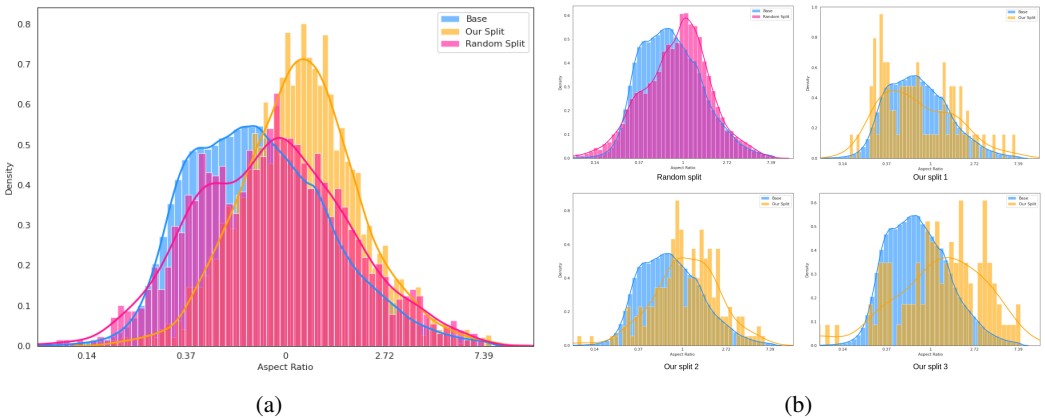

(a)                                                    (b)

Figure 2: The choice of base and novel class split can have a strong effect on the distribution of box aspect ratios. This histogram on COCO (a) and LVIS (b) shows: (blue) the distribution of aspect ratios for base classes; (pink) a randomly selected set of novel classes — note the *bimodal* structure, suggesting that different classes might have quite different bounding box distributions; and (yellow) for each of our selected hard set of novel classes in COCO-ARShift and LVIS-ARShift. Note that these classes are *"rare"* classes in LVIS so the histogram appears sparse. Experimental evidence (Tables 1, 2 and 3) suggests that a standard RPN trained on the base set performs poorly on our novel classes, while our CoRPN's address the issue and substantially improve the performance. We believe that our hard splits in ARShift are relatively easy to generate and examples are provided in the appendix.

| Method | shot=1 | 2 | 3 | 5 | 10 |
|---|---|---|---|---|---|
| TFA w/ cos + CoRPN's | **28.1** | **31.7** | **36.0** | **38.6** | **44.7** |
| DeFRCN + CoRPN's | **37.0** | **49.3** | **55.1** | **56.7** | **59.0** |
| TFA w/ cos (Wang et al., 2020) | 19.1 | 27.2 | 28.0 | 34.4 | 42.1 |
| DeFRCN (Qiao et al., 2021) | 31.0 | 44.6 | 47.5 | 55.2 | 57.2 |
| DeFRCN Ensemble of Experts | 29.7 | 42.7 | 50.9 | 55.3 | 56.0 |

Table 1: Our proposed split VOC-ARShift causes serious problems for current state-of-the-art few-shot detectors, which are resolved by using CoRPN's. CoRPN's bring significant improvements (AP50) to all baseline models under distribution shift of base/novel class box aspect ratios. Note that naïve ensemble of RPN experts sometimes leads to even worse results, potentially due to the overspecialization of its individual RPNs. Results in **bold** are the better result between ours and the baselines. Results in **red** are the best.

for novel classes are randomly initialized, making the classifier a $(|C_b| + |C_n|)$-way classifier. In phase 2, the model is fine-tuned using either a set of few novel class instances or a balanced dataset containing both novel and base classes. The classifier sees the ground truth box and RPN boxes; it is typically trained to regard RPN boxes with IoU$\geq$0.7 as positive, and with IoU$<$0.3 as negative. After the fine-tuning phase, we evaluate our model by average precision (AP) on novel and base categories. Although the focus of few-shot detection is the novel classes, since most test images contain instances from both base and novel classes, it is essential to maintain good performance on base classes.

We adopt the widely-used Faster R-CNN (Ren et al., 2015) as our base model. Faster R-CNN is a two-stage detector, which consists of a backbone image feature extractor, an RPN, followed by the region of interest (ROI) pooling layer, and a bounding box classifier and a bounding box regressor on top of the model. The RPN determines if a box is a foreground or a background box. Following the RPN is non-maximum suppression (NMS) which ranks and selects top proposal boxes. In phase 1, the whole model is trained on many-shot base class instances. Phase 2 fine-tunes part of the model on novel class instances with other parts frozen. Specifically, for TFA (Wang et al., 2020), only the top layer of the bounding box classifier and regressor are fine-tuned. For DeFRCN (Qiao et al., 2021), the whole model is fine-tuned except for the convolutions in the bounding box classifier and regressor.

## 3.2  PILOT STUDY: BOX DISTRIBUTION SHIFT

In the real world, the bounding box distribution of these novel categories often differs from the base categories, resulting in unfair few-shot performances. Namely, the difference in the distribution of

|  | Method | 1-shot | | | 2-shot | | | 3-shot | | |
|---|---|---|---|---|---|---|---|---|---|---|
|  |  | AP | AP50 | AP75 | AP | AP50 | AP75 | AP | AP50 | AP75 |
| Ours | DeFRCN + CoRPN's | **8.7** | **15.3** | **8.7** | **12.4** | 20.5 | **13.4** | **14.7** | **24.4** | **15.9** |
| Baselines | DeFRCN (Qiao et al., 2021) | 7.1 | 13.0 | 7.0 | 11.5 | 20.3 | 12.5 | 14.3 | **24.4** | 15.1 |
|  | DeFRCN Ensemble of Experts | 8.0 | 14.3 | 7.9 | 12.3 | **20.6** | 13.0 | 14.2 | 23.5 | 15.0 |

Table 2: Our proposed split COCO-ARShift causes serious problems for current state-of-the-art few-shot detectors, which are resolved by using CoRPN's. CoRPN's consistently outperform the DeFRCN (Qiao et al., 2021) baseline in all cases (novel split in 1, 2 and 3 shots) and are superior to the ensemble of experts method overall. Results in **red** are the best.

|  | Method | Split 1 | | | Split 2 | | | Split 3 | | | Entire test set | | |
|---|---|---|---|---|---|---|---|---|---|---|---|---|---|
|  |  | AP | AP50 | AP75 | AP | AP50 | AP75 | AP | AP50 | AP75 | AP | AP50 | AP75 |
| Ours | DeFRCN + CoRPN's | **12.3** | **31.0** | **9.4** | **16.0** | **27.5** | **18.8** | **13.7** | **20.4** | **16.8** | **15.3** | 26.4 | **15.6** |
| Baseline | DeFRCN (Qiao et al., 2021) | 9.6 | 22.9 | 7.0 | 8.0 | 19.1 | 1.7 | 9.0 | 18.8 | 6.3 | 15.0 | **26.6** | 14.2 |
|  | DeFRCN Ensemble of Experts | 11.5 | 26.0 | 8.4 | 14.8 | 24.3 | 12.1 | 12.4 | 20.1 | 14.7 | 14.8 | 25.0 | 13.9 |

Table 3: Our proposed three splits on LVIS (LVIS-ARShift) cause serious problems for current state-of-the-art few-shot detectors, which are resolved by using CoRPN's. As a reference, we also provide the results on all rare classes in the entire LVIS test set. Our CoRPN's outperform the baseline by large margins on all the splits with aspect ratio distribution shift, especially in AP75, suggesting that CoRPN's improve the quality of detection under aspect ratio shift. CoRPN's also marginally outperform the baseline on the entire test set in mAP and AP75. Results in **red** are the best.

box scale, aspect ratio, and center location all inhibit a successful transfer. Previous work tries to alleviate the scale issues with multiscale features and the location issues with translation invariance of convolution. However, these approaches fail to solve all of these problems, especially when the distribution between base and novel class box *aspect ratios* has a significant shift. We simulate this scenario by proposing new splits: our ARShift benchmark, on the PASCAL VOC, COCO and LVIS datasets that emphasize this distribution shift and the fairness to different aspect ratio distributions.

We manually select a set of classes that will likely have a different box distribution to the base categories. Figure 2 shows that our split has a more significant shift in the box aspect ratio distribution. As an naïve approach to alleviating this issue, we modified the RPN classifier to be an ensemble of expert RPN classifiers. Instead of using one RPN classifier for all proposals, we use 3 RPN classifiers for the anchors of 3 different aspect ratios (0.5, 1, and 2.0 respectively). The 3 RPN classifiers independently output their prediction for their respective anchors, which are combined as the final prediction. Tables 1, 2 and 3 show that compared to the baseline *DeFRCN* (Qiao et al., 2021), this approach represented by '*DeFRCN Ensemble of Experts*' performs comparably and thus cannot fully address the issue. Intuitively, in this ensemble of experts method, individual RPN classifiers might be overspecializing and so facing generalization problems. Instead, we propose a method where we do not explicitly enforce each RPN classifier to specialize in an aspect ratio, but let the *specialization emerge in the learning process*. This method, named *CoRPN's*, shows a large improvement over the baselines in Tables 1, 2 and 3.

### 3.3 Learning Cooperating RPN's (CoRPN's)

We would like to alter the RPN to improve the population of boxes reported, especially on our ARShift benchmark where novel classes has a large box distribution shift. We expect that doing so affects mainly the few-shot case. As illustrated in Figure 3, we use multiple redundant RPN classifiers, but our goals imply that these RPN classifiers need to be trained to cooperate (*i.e.*, they should not be a pure ensemble of experts). In what follows, we use the term *RPN* and *RPN classifier* interchangeably unless otherwise noted. In particular, we train and evaluate our RPN classifiers using an OR strategy – a box is classified with the label reported by the most confident RPN, which gets the gradient during training. This has two effects. First, the RPN's can specialize to a degree, though we do not allow the RPN's to drift too far apart. In turn, if one RPN misses a positive box, the other might find it. Second, the training strategy may improve the variation of proposals, which is especially essential when dealing with a different proposal box distribution in the few-shot case. Both effects may bring improvements to the model's fairness with respect to aspect ratio distribution.

Faster R-CNN's RPN consists of a feature extractor, a binary classifier (which decides whether a box is foreground or background), and a bounding box regressor (which is not relevant to our current purpose). We do not intend for our RPN's to use distinct sets of features, since it would introduce a large number of additional parameters, so we construct redundant classifiers while keeping both the feature extractor and the bounding box regressor shared between all RPN's.

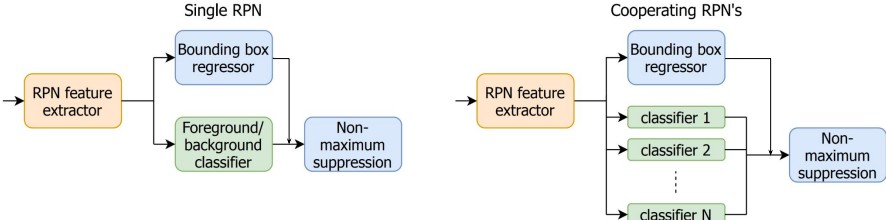

Figure 3: Illustration for our CoRPN's network. **Left** is the original structure for the RPN in Faster R-CNN; **right** is our CoRPN's, consisting of cooperating bounding box classifiers. Note that **we only add multiple classifier heads that share features**, so the model complexity of CoRPN's does not significantly exceeds a regular RPN. For convenience, we use the term ***RPN*** and ***RPN classifier*** **interchangeably**.

An RPN with a single classifier is trained with a cross-entropy loss $\mathcal{L}_{\text{cls}} = \mathcal{L}_{\text{CE}}$ and produces a single prediction. In our case, we train $N$ different binary classifiers simultaneously, and must determine (1) what prediction is made at test time and (2) what gradient goes to what classifier at training time. At test time, a given box gets the score from the most confident RPN. If the highest foreground probability is closer to one than the highest background probability, the box is predicted to be foreground; otherwise, it is predicted to be the background. In training time, merely taking the gradient of the best RPN score is not good enough, because the model may collapse to the trivial solution where one RPN scores all boxes, and the others do nothing interesting. For any foreground box, we want at least one RPN to have a very confident foreground prediction *and* all others to have good foreground scores too (so that no foreground box is missed).

We use the following strategy. For a specific anchor box $i$, each RPN $j$ (of the $N$ RPN's) outputs a raw score $r_i^j$, indicating if the box is a foreground box or not: $r_i = [r_i^1, r_i^2, \ldots, r_i^N]$. After applying a sigmoid, the $j$th RPN produces the foreground probability $f_i^j = \sigma(r_i^j)$ for anchor box $i$. We choose the score from the $j^*$th RPN such that

$$j^* = \operatorname{argmin}_j \min\{f_i^j, 1 - f_i^j\}, \tag{1}$$

namely the most certain RPN which produces probability closest to the edge of the $[0, 1]$ interval. At training time, only the chosen $j^*$th RPN gets the gradient from anchor box $i$. The RPN selection procedure is per-box, and even adjacent boxes can pass through different RPN's.

Other than the standard cross-entropy loss, we use two additional loss terms: a diversity loss $L_{\text{div}}$ encourages RPN's to be distinct, and a cooperation loss $L_{\text{coop}}$ encourages cooperation and suppresses foreground false negatives. The final loss is

$$\mathcal{L}_{\text{cls}} := \mathcal{L}_{\text{CE}}^{j^*} + \lambda_d \mathcal{L}_{\text{div}} + \lambda_c \mathcal{L}_{\text{coop}}, \tag{2}$$

where $\lambda_d$ and $\lambda_c$ are trade-off hyperparameters.

### 3.4 Enforcing Diversity

We do not want our RPN's to be too similar and prefer their specialization. For each positive anchor box, RPN responses should be different because we want different RPN's to cover different types of proposal boxes. To this end, we propose a loss function to enforce diversity among RPN's. Given a set of $N_A$ anchor boxes, the $N$ RPN's produce an $N$ by $N_A$ matrix of probabilities $\mathcal{F} = [f^1, f^2, \ldots, f^N]^T$. The covariance matrix is

$$\Sigma_{jk}(\mathcal{F}) = \mathrm{E}[(f^j - \mathrm{E}[f^j])(f^k - \mathrm{E}[f^k])^T]. \tag{3}$$

We define the diversity loss $\mathcal{L}_{\text{div}}$ by the log determinant loss

$$\mathcal{L}_{\text{div}} := -\log(\det(\Sigma(\mathcal{F}))). \tag{4}$$

This log determinant loss has been widely used in previous work (Boyd & Vandenberghe, 2008; Dhillon, 2008) to encourage diversity. By this diversity loss, we encourage the probability matrix to have rank $N$, so each RPN is reacting differently on the collection of $N_A$ boxes. This procedure ensures each RPN to be the most certain RPN for some boxes, so that every RPN is being selected and trained. Omitting this loss can cause some RPN classifier to receive little training.

| | Novel Class | | | | | | | | | Base Class | | | | | | | | |
| Method | 1-shot | | | 2-shot | | | 3-shot | | | 1-shot | | | 2-shot | | | 3-shot | | |
| | AP | AP50 | AP75 | AP | AP50 | AP75 | AP | AP50 | AP75 | AP | AP50 | AP75 | AP | AP50 | AP75 | AP | AP50 | AP75 |
|---|---|---|---|---|---|---|---|---|---|---|---|---|---|---|---|---|---|---|
| TFA w/ cos + CoRPN's | **4.1** | **7.2** | **4.4** | **5.4** | 9.6 | **5.6** | **7.1** | **13.2** | **7.2** | **34.1** | **55.1** | **36.5** | **34.7** | **55.3** | 37.3 | **34.8** | **55.2** | 37.6 |
| DeFRCN + CoRPN's | **5.0** | **9.7** | **4.8** | **8.6** | 16.1 | **7.8** | **10.9** | **20.0** | **10.4** | **30.3** | **45.6** | **33.2** | **31.2** | **47.9** | 34.2 | **32.0** | **48.1** | **35.1** |
| TFA w/ cos (Wang et al., 2020) | 3.4 | 5.8 | 3.8 | 4.6 | 8.3 | 4.8 | 6.6 | 12.1 | 6.5 | 34.1 | 54.7 | 36.4 | 34.7 | 55.1 | **37.6** | 34.7 | 54.8 | **37.9** |
| DeFRCN (Qiao et al., 2021) | 4.8 | 9.5 | 4.4 | 8.5 | **16.3** | 7.8 | 10.7 | **20.0** | 10.3 | 30.3 | 45.7 | 33.4 | 31.2 | 47.1 | **34.5** | 31.8 | 47.9 | 35.1 |
| MPSR** (Wu et al., 2020) | 2.3 | 4.1 | 2.3 | 3.5 | 6.3 | 3.4 | 5.2 | 9.5 | 5.1 | 12.1 | 17.1 | 14.2 | 14.4 | 20.7 | 16.9 | 15.8 | 23.3 | 18.3 |
| FsDetView (Xiao & Marlet, 2020) | 3.2 | 8.9 | 1.4 | 4.9 | 13.3 | 2.3 | 6.7 | 18.6 | 2.9 | 2.4 | 7.0 | 1.0 | 4.4 | 11.9 | 2.2 | 4.9 | 13.6 | 2.2 |
| FSOD* (Fan et al., 2020) | 2.4 | 4.8 | 2.0 | 2.9 | 5.9 | 2.7 | 3.7 | 7.2 | 3.3 | 11.9 | 20.3 | 12.5 | 15.6 | 24.4 | 17.2 | 17.4 | 27.3 | 19.0 |

Table 4: CoRPN's not only improve performance on our hard split, but also perform comparably with state of the art on more traditional splits without forgetting the base classes, hence improving model fairness for aspect ratio distribution. CoRPN's mostly beat strong baselines in few-shot detection performance on the (extremely difficult) COCO novel classes task for 1, 2, and 3-shot cases. Results in **bold** are the better result between ours and the baselines. All approaches are evaluated following the standard procedure in Xiao & Marlet (2020). *Model re-evaluated using the standard procedure (with base and novel classes joint space) for a fair comparison. **Model evaluated using public code and pre-trained base classes detector.

| Ratio | Novel Set 1 | Novel Set 2 | Novel Set 3 |
|---|---|---|---|
| 0.5 | RPN2 99.68% | RPN5 99.72% | RPN2 98.64% |
| 1.0 | RPN3 99.68% | RPN2 99.67% | RPN2 99.83% |
| 2.0 | RPN1 99.67% | RPN1 99.59% | RPN1 10.75%, RPN2 89.25% |

Table 5: Aspect ratio coverage of different RPN classifiers on VOC standard splits. For example, *RPN1* refers to the first RPN classifier in the CoRPN's. This provides evidence that CoRPN's indeed learn to specialize in box aspect ratios, despite no direct supervision.

## 3.5 LEARNING TO COOPERATE

We also want the RPN's to cooperate so that they all agree to a certain extent for foreground boxes. We propose a cooperation loss to prevent any RPN from firmly rejecting any foreground box. For foreground box $i$, with the $j$th RPN, we define the cooperation loss

$$\mathcal{L}_{\text{coop}}^{i,j} := \max\{0, \phi - f_i^j\}, \tag{5}$$

where $\phi$ is a constant parameter (usually less than 0.5), acting as a lower bound for each RPN's probability assigning to a foreground box. If an RPN's response is below $\phi$, that RPN is going to be penalized. The final cooperation loss is an average of cooperation losses over all foreground boxes and all RPN's.

## 4 EXPERIMENTS

**Benchmarks.** We propose a new base/novel split on both the PASCAL VOC (Everingham et al., 2010), MS-COCO (Lin et al., 2014) and LVIS (Gupta et al., 2019) datasets to simulate the real-world scenario, where the novel class box distribution deviates from the base class counterpart. Our proposed split VOC-ARShift is similar to the conventional few-shot detection VOC split (Kang et al., 2019; Wang et al., 2020). We use the images and annotations of VOC (07 + 12) and select 15 classes as base classes, and leave the rest as novel classes. In our proposed COCO-ARShift split, we use the images and annotation of COCO 2014, select the 20 VOC classes as training, and select 10 out of the other 60 classes as novel classes. We select these classes to explicitly produce a distribution shift in the box aspect ratios, as shown in Figure 2. In our LVIS-ARShift benchmark, we use LVIS v0.5 and 10 shots following Wang et al. (2020). In each setting, the base classes are the 20 VOC classes in COCO, while the novel classes are 10 rare classes manually picked that have an aspect ratio distribution shift from the base classes. The detailed classes of our proposed splits are in Section H in the appendix.

Apart from our proposed ARShift setting, we also evaluate on two widely-used few-shot detection benchmarks (Kang et al., 2019; Wang et al., 2020) based on PASCAL VOC and COCO. For a fair comparison, we use the same train/test splits and novel class instances as in Kang et al. (2019); Wang et al. (2020) to train and evaluate all models. On COCO, we report base/novel classes AP, AP50, and AP75 under shots 1, 2, 3, 5, 10, and 30. On PASCAL VOC, we report AP50 for three different base/novel class splits under shots 1, 2, 3, 5, and 10. Following Wang et al. (2020) and Qiao et al. (2021), we use Faster R-CNN as our base model and use an ImageNet pre-trained (Russakovsky et al., 2015) ResNet-101 as the backbone, unless otherwise noted.

**Training Procedure.** Our training and fine-tuning procedures are consistent with previous work TFA (Wang et al., 2020) and DeFRCN (Qiao et al., 2021). On PASCAL VOC, at phase 1 base

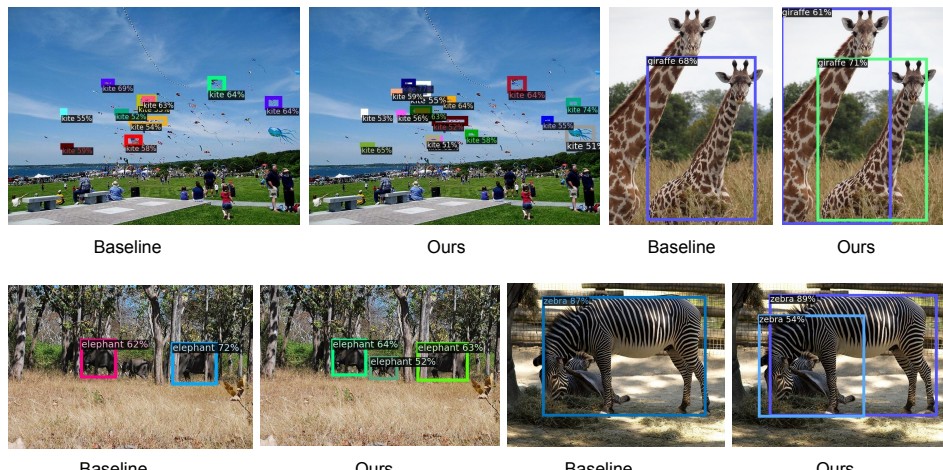

Figure 4: Qualitative results of our CoRPN's + DeFRCN on COCO-ARShift on 5-shots, compared with the DeFRCN baseline. The results illustrate that our approach discovers novel objects missed by the baseline, and also has less misclassification.

classes training, each model is trained on the union set of VOC 07+12 trainval data. Evaluation is on the VOC 07 test set. At the fine-tuning phase, each model is fine-tuned with a balanced few-shot dataset sampled from VOC 07+12 that contains both base classes and novel classes. On COCO, following Wang et al. (2020), the fine-tuning phase is two-stage: at stage 1, we fine-tune the model on novel classes; at stage 2, we then fine-tune the model with a balanced few-shot dataset containing both base and novel classes. Please refer to Section C of the appendix for our implementation details and hyperparameters.

**Baselines and Evaluation Procedure.** We mainly focus on comparing against the state-of-the-art baseline DeFRCN (Qiao et al., 2021), and a widely-used previous work TFA (Wang et al., 2020). Our approach incorporates CoRPN's into the baseline models, while keeping other model components and design choices unchanged. In addition, we thoroughly compare with a variety of recent few-shot detectors, including FSOD (Fan et al., 2020), MPSR (Wu et al., 2020), FSDetView (Xiao & Marlet, 2020). These baselines address other aspects of few-shot detection which are different from us (Section 2), and their modifications are thus *largely orthogonal to our effort* here. Note that our evaluation follows the standard procedure in Wang et al. (2020). This standard procedure computes AP separately for novel and base categories for a detector that is engineered to detect both novel and base classes (($|C_b| + |C_n|$)-way). We focus on the novel class performance, and also report the base class performance in Table 4. For work (Fan et al., 2020) with different evaluation procedures, we re-evaluate their methods with the standard procedure, so that results in Tables 4 can be different from the original reported results.

## 4.1 MAIN RESULTS

Our evaluation mainly focuses on our proposed ARShift splits, but also includes the conventional splits. Also, we focus on the *extremely few-shot regime*, which is the most challenging scenario for few-shot detection. Tables 1, 2 and 3 show the detection performance on our proposed ARShift splits where a bounding box aspect ratio distribution shift is present between base and novel classes. Table 4 summarize the detection results for base and novel classes in low shots on the conventional COCO benchmark, respectively. For completeness, the results for base and novel classes in higher shots on PASCAL VOC and COCO are summarized in Section B of the appendix, where our model also performs comparably. We present our qualitative results in Figures 4 and 6.

**CoRPN's consistently outperform the baselines.** Tables 1, 2 and 3 show that CoRPN's greatly outperform the baseline TFA (Wang et al., 2020), DeFRCN (Qiao et al., 2021) and an aspect ratio ensemble modified from the DeFRCN model, especially in very low shots. Especially, we provide the results on three additional base and novel class splits on the LVIS dataset (Gupta et al., 2019) in Table 3. LVIS is a much more challenging long-tail recognition dataset, containing a large amount of rare classes, hence the fairness for aspect ratio is more crucial. We also provide the results on all rare classes in the LVIS test set as a reference. Our CoRPN's outperform the state-of-the-art

| Method | | AP | AP50 | AP75 |
|---|---|---|---|---|
| DeFRCN Qiao et al. (2021) | | 13.6 | 31.0 | 9.6 |
| CoRPN's | | **18.8** | **37.0** | **17.2** |
| No Cooperation Loss | | 14.7 | 30.9 | 11.7 |
| No Diversity Loss | | 13.8 | 30.2 | 10.0 |
| | $\phi = 0.1$ | 14.2 | 31.0 | 11.1 |
| Threshold | $\phi = 0.4$ | 16.8 | 35.2 | 13.5 |
| | $\phi = 0.7$ | 14.7 | 31.5 | 11.6 |
| | $\phi = 0.9$ | 14.8 | 33.6 | 10.1 |

Table 6: Our diversity loss and cooperation loss are both required for CoRPN's to obtain the largest improvement. A sub-optimal threshold in the cooperation loss also has an adverse effect on performance. The table shows 1-shot novel class performance of all models under our proposed VOC-ARShift novel split.

DeFRCN baseline by large margins on all the splits with aspect ratio distribution shift, especially in AP75, suggesting that CoRPN's improve the quality of detection in such scenarios. CoRPN's also marginally outperform the DeFRCN baseline on the entire set of rare classes in mAP and AP75 on LVIS. On the conventional splits, as shown in Tables 4, CoRPN's *consistently improve over TFA for all shots*, and also marginally outperform DeFRCN on the challenging COCO dataset. The combination of both results shows a significant improvement in aspect ratio distribution fairness in our model.

**CoRPN's beat other state of the art.** With our simple modification on RPN, we also outperform other sophisticated approaches in the very low-shot regime on the more challenging COCO dataset. In particular, we significantly outperform baselines that introduce attention mechanisms for adjusting proposal generation (Hsieh et al., 2019; Fan et al., 2020) under the standard evaluation procedure. We believe CoRPN's could be combined with other approaches with improvements from different perspectives, such as exploiting better multi-scale representation (Wu et al., 2020), incorporating metric learning (Yang et al., 2020), or adding feature aggregation module (Xiao & Marlet, 2020) for further improvements.

**CoRPN's don't forget base classes.** While improving detection on novel classes through fine-tuning, we maintain strong performance on base classes – *there is no catastrophic forgetting* (Table 4). By contrast, the base class performance of some state-of-the-art baselines dramatically drops, demonstrating that they cannot fairly detect novel and base classes.

## 4.2 ABLATION STUDY

We investigate how the proposals of CoRPN's change and conduct a series of ablations that evaluate the contribution of each loss component and different design choices. Specifically, we find that: (1) CoRPN's specialize in different aspect ratios without explicit supervision; (2) Our cooperation loss and diversity loss are both necessary for CoRPN's to improve fairness; (3) (in Section F of the appendix) CoRPN's outperform other baselines such as with larger RPN sub-networks, an existing cosine loss based diversity, and bootstrapping.

**Specialization for aspect ratios emerges in CoRPN's training.** Table 5 shows that the boxes of three different aspect ratios are handled by different RPN classifiers in CoRPN's. Instead of explicitly training different RPN classifiers to handle different aspect ratios, CoRPN's learn a more flexible specialization, improving the performance.

**CoRPN's need both diversity and cooperation losses.** Table 6 shows that after removing either loss, CoRPN's does not improve the performance over the baseline. Also, when the threshold hyper-parameter is suboptimal, the performance surpasses the baseline but still substantially underperforms the CoRPN's with the optimal hyperparameter.

## 5 CONCLUSION

We identify the bias of few-shot detectors towards familiar aspect ratio distribution. As illustrated in our ARShift benchmark, a substantial improvement on under-represented aspect ratio distribution can be obtained by our proposed CoRPN's which produce more informative proposals. Our method achieves a new state of the art on both our proposed settings with hard base and novel splits and widely-used benchmarks in the very few-shot regime. This is achieved by training CoRPN's with diversity and cooperation losses.

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

## A  APPENDIX

*This appendix provides additional experimental results and details that support the approach in the main paper and are not included there due to limited space. The seven sections include (1) additional results for higher-shots on the COCO-ARShift, comparisons with additional baselines for ARShift, and results on base classes before and after finetuning; (2) explainations for additional evaluation and experimental details; (3) discussion of the training and inference time and memory; (4) results on conventional splits; (5) analysis with additional ablation study; (6) addtional qualitative results; and (7) details of the proposed base and novel class splits on ARShift for PASCAL VOC, COCO, and LVIS.*

## B  ADDITIONAL RESULTS

### Higher-Shot Results on COCO-ARShift

In Table 2 of the main paper, we show that CoRPN's outperform the DeFRCN (Qiao et al., 2021) baseline consistently on 1, 2, and 3 shots. Table 10 shows that this performance improvement persists in 5, 10, and 30 shots. In this scenario where more support instances are available, CoRPN's consistently outperform the DeFRCN (Qiao et al., 2021) baseline.

### Additional Baseline on ARShift

In Table 1 of the main paper, we mainly compare our method against the state-of-the-art method DeFRCN (Qiao et al., 2021) on our VOC-ARShift. In Table 11 we also evaluate an additional baseline FSCE (Sun et al., 2021). Note that other recently published methods represented by FSCE underperform our DeFRCN baseline by significant margins on our ARShift splits as well, so we do not include these other methods in our main paper.

### Results on Base classes

Here we provide the detection results on base classes after base classes training (stage 1) and after fine-tuning (stage 2). As shown in Table 9, the performance of our CoRPN's + DeFRCN on base classes is comparable with DeFRCN (Qiao et al., 2021), while we achieve large improvements on novel test classes as shown in Table 1 of the main paper. After fine-tuning, our CoRPN's also do not forget base classes. Table 9 shows that the performance of our CoRPN's + DeFRCN after fine-tuning is still comparable with the baseline.

### Average Recall

We present the average recall (AR) result on the three splits in our LVIS-ARShift benchmark in Table 7. Here we show AR1000 by the convention of COCO. Our CoRPN's also improve few-shot detection in the AR, meaning that our CoRPN's miss fewer novel objects under aspect ratio distribution shift.

### Per-Category Result

In Table 8 we present the 1-shot AP50 for each novel class on our VOC-ARShift benchmark. Our CoRPN's improve upon the vanilla DeFRCN baseline by large amounts in 4 of the 5 categories, while the naive ensemble of experts only achieves marginal improvements.

### Analysis of Proposal Aspect Ratio

Figure 5 shows the aspect ratio distribution of proposals from the baseline RPN and our CoRPN's. Our CoRPN's produce more diverse proposals that are robust to the aspect ratio distribution shift.

## C  IMPLEMENTATION AND EVALUATION DETAILS

**Implementation Details and Hyperparameters.** For ease of comparison, we use the same values for all shared training and fine-tuning hyperparameters (batch size, learning rate, momentum, weight decay, *etc.*) as the baselines Wang et al. (2020) and Qiao et al. (2021). CoRPN's have the following additional hyperparameters: the number of RPN's, the cooperation loss threshold $\phi$, the diversity loss trade-off $\lambda_d$, and the cooperation loss trade-off $\lambda_c$. For COCO and LVIS, we directly used

| | Method | Split 1 | Split 2 | Split 3 |
|---|---|---|---|---|
| Ours | DeFRCN + CoRPN's | **18.4** | **24.8** | **28.3** |
| Baselines | DeFRCN (Qiao et al., 2021) | 17.6 | 18.5 | 13.9 |
| | DeFRCN Ensemble of Experts | 18.1 | 23.4 | 25.0 |

Table 7: Average recall (AR1000) on our LVIS-ARShift benchmark. We achieve performance improvements over both baselines in recall.

| | Method | Train | Bottle | Aeroplane | Horse | Bus |
|---|---|---|---|---|---|---|
| Ours | DeFRCN + CoRPN's | **46.2** | **4.5** | **43.4** | 10.5 | **80.4** |
| Baselines | DeFRCN (Qiao et al., 2021) | 27.5 | 1.8 | 23.2 | 50.6 | 51.7 |
| | DeFRCN Ensemble of Experts | 27.6 | 3.1 | 24.3 | **54.7** | 38.6 |

Table 8: Per-Category 1-shot AP50 result on our VOC-ARShift benchmark. Our method achieves large improvement in 4 of the 5 classes.

hyperparameter sets that worked well on PASCAL VOC. In Table 4, we report CoRPN's detection results. We find that hyperparameters selected from PASCAL VOC are *generalizable* to the more challenging COCO and LVIS benchmark. Results reported in Table 4 are obtained with 5 RPNs, $\phi$ = 0.3, $\lambda_c$ = 1, $\lambda_d$ = 0.025 for TFA(Wang et al., 2020), and $\lambda_c$ = 2, $\lambda_d$ = 0.05 for DeFRCN (Qiao et al., 2021).

**Selection Procedure – Cumulative Variance.** We summarized the hyperparameter selection criteria for PASCAL VOC in the experiment section in the main paper. The second criterion is the cumulative variance in the RPN's response matrix to foreground boxes. Specifically, given a set of $M$ anchor boxes, the $N$ RPN's produce an $N$ by $M$ matrix of probabilities $\mathcal{F} = [f^1, f^2, \ldots, f^N]^T$. We run a principal component analysis (PCA) on $\mathcal{F}$ with $N$ components and compare the cumulative percentage of variance explained by each component. We would like the variance to be distributed across components.

**Evaluation.** As mentioned in the main paper, for a fair comparison, we use the *standard evaluation procedure* for all the compared models. We also compare against other approaches with the same novel class instances and test images. Authors of methods compared in this submission have helped us ensure the performance we report is a proper reflection of their methods, and we will acknowledge properly in any final version. In the standard evaluation procedure, when a test image comes in, the model has no assumption on what category the image contains (Wang et al., 2020). The detector's classifier is $(|C_b|+|C_n|)$-way, detecting objects from a joint space of both base and novel categories. In the main paper, we marked results with * if they were *re*-evaluated under the standard procedure, and with ** if the original work used the standard procedure, but the results were not reported and so they were evaluated by us.

Specifically, below we include the details on how we obtained the results (with special marks) in the main paper; other results (without special marks) (Wang et al., 2020; Pérez-Rúa et al., 2020; Yang et al., 2020; Sun et al., 2021; Li et al., 2021a; Zhu et al., 2021) including the concurrent work (Sun et al., 2021; Li et al., 2021a; Zhu et al., 2021) are from the original papers.

- We fine-tuned and evaluated MPSR using the publicly released code and the pre-trained detection model for base classes (Wu et al., 2020).

- We re-evaluated FSOD (Fan et al., 2020) and CoAE (Hsieh et al., 2019) under the standard procedure, using the publicly released code and the pre-trained model (Fan et al., 2020; Hsieh et al., 2019). FSOD uses a class-agnostic 2-way classifier that determines if an object is foreground or background. At inference time, FSOD produces a balanced number of proposals per novel category and collects these proposals for NMS. We adapt FSOD under the standard procedure such that there are a balanced number of proposals for all base and novel categories. For a fair comparison, we also fine-tuned FSOD with the same novel category instance(s) as in TFA (Wang et al., 2020).

- We re-evaluated CoAE under the standard evaluation, using the publicly released code and a pre-trained model (Hsieh et al., 2019). For each test image containing a certain category, CoAE samples support image(s) from this category and collects boxes based on

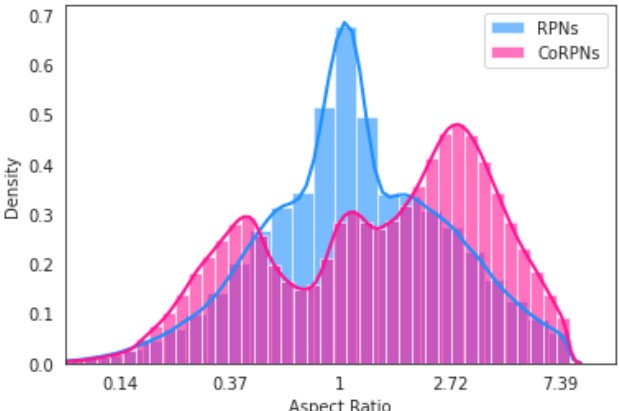

Figure 5: The aspect ratio distribution of proposals from the baseline RPN and our CoRPN's. Our CoRPN's produce more diverse proposals robust to the aspect ratio distribution shift.

| Method | After Phase 1 w/o fine-tuning | After Phase 2 1-shot | 2-shot | 3-shot | 5-shot | 10-shot |
|---|---|---|---|---|---|---|
| DeFRCN (Qiao et al., 2021) | 81.4 | 76.5 | 77.0 | 76.9 | 77.0 | 76.7 |
| DeFRCN + CoRPN's (Ours) | 82.8 | 75.6 | 75.8 | 75.7 | 75.5 | 75.4 |

Table 9: Base classes AP50 on VOC-ARShift after base class training (phase 1) and after fine-tuning (phase 2). The same hyperparameter setting applies to both models. Notice that ours is higher after phase 1 and slightly lower but comparable with the results of DeFRCN after phase 2, but our method achieves significantly better performance on test classes as shown in Table 1 of the main paper (e.g., 7.3AP on 1-shot and 6.6 AP on 2-shot). Neither our method nor the baseline forgets base classes significantly.

the support image(s). At inference time, instead of providing each test image with support feature(s) from this category, we provide each test image support feature(s) from all base and novel categories. We then collect boxes from all categories and evaluate them. For a fair comparison, we also fine-tuned CoAE with the same novel category instance(s) as in TFA (Wang et al., 2020).

## D   TRAINING & INFERENCE TIME AND MEMORY

The only architectural difference between CoRPN's and the baseline RPNs of TFA (Wang et al., 2020) and DeFRCN(Qiao et al., 2021) is that CoRPN's have multiple RPN classifiers. Note that we only duplicate the RPN classifiers, but not the entire RPN. In the baselines, the RPN's classifier is a 1×1 convolutional layer, with input channels as the number of feature channels (256) and output channels as the number of cell anchors (3). Compared with the baselines, CoRPN's with 5 RPN's have four additional RPN classifiers and thus consist of 256×12 additional parameters. In our experiments, we find that CoRPN's with 5 RPN's increase the training time *by only 3%*, with roughly the same inference time and same memory footprint, compared with the baselines.

## E   RESULTS ON CONVENTIONAL SPLITS

In Table 4 of the main paper, we show that CoRPN's perform comparably or even better on conventional splits on 1, 2, and 3 shots on COCO. In this section we present more conventional split results. Table 12 shows PASCAL VOC conventional split results where our CoRPN's perform comparably with the baselines. Table 13 shows the results on COCO 5, 10, and 30 shots, where our method also obtains similar performance to the baseline.

| Method | 5-shot | | | 10-shot | | | 30-shot | | |
|---|---|---|---|---|---|---|---|---|---|
| | AP | AP50 | AP75 | AP | AP50 | AP75 | AP | AP50 | AP75 |
| DeFRCN (Qiao et al., 2021) | 16.9 | **27.4** | 18.0 | 18.9 | 30.8 | 20.2 | 21.2 | 34.5 | **22.2** |
| DeFRCN + CoRPN's (Ours) | **17.2** | 27.0 | **18.3** | **19.3** | **31.0** | **21.0** | **21.4** | **34.9** | 22.0 |

Table 10: When there are many shots (our proposed COCO-ARShift in 5, 10 and 30 shots), while the impact of using CoRPN's is reduced, CoRPN's still outperform the DeFRCN (Qiao et al., 2021) baseline in most cases. The reduced improvement is likely because missing objects in the RPN stage becomes a less important effect. Results in **red** are the best.

| Method | shot=1 | 2 | 3 | 5 | 10 |
|---|---|---|---|---|---|
| TFA w/ cos + CoRPN's | **28.1** | **31.7** | **36.0** | **38.6** | **44.7** |
| DeFRCN + CoRPN's | **37.0** | **49.3** | **55.1** | **56.7** | **59.0** |
| TFA w/ cos (Wang et al., 2020) | 19.1 | 27.2 | 28.0 | 34.4 | 42.1 |
| FSCE (Sun et al., 2021) | 23.7 | 33.5 | 35.8 | 41.5 | 48.1 |
| DeFRCN (Qiao et al., 2021) | 31.0 | 44.6 | 47.5 | 55.2 | 57.2 |
| DeFRCN Ensemble | 29.7 | 42.7 | 50.9 | 55.3 | 56.0 |

Table 11: In Table 1 of the main paper, we mainly compare our method against the state-of-the-art method DeFRCN (Qiao et al., 2021) on our VOC-ARShift. Here we evaluate the result of an additional baseline FSCE (Sun et al., 2021) (marked in green). Note that other recently published methods represented by FSCE underperform our DeFRCN baseline by significant margins on our ARShift splits as well, so we do not include these methods in our main paper.

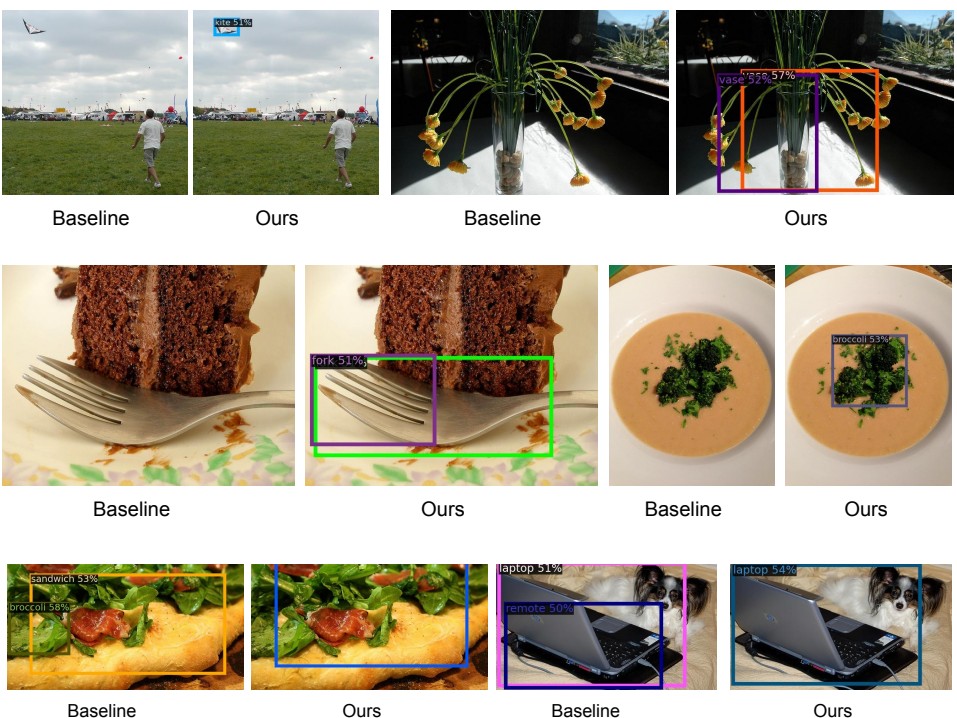

Figure 6: Additional qualitative results of our CoRPN's + DeFRCN on COCO-ARShift on 5-shots, compared with the DeFRCN baseline.

# F   ADDITIONAL ANALYSIS

We present some additional ablation studies for our CoRPN's. Note that these ablations are performed on the conventional split. Even on conventional splits, our CoRPN's outperform these naive strategies. Our CoRPN's will likely achieve more performance improvement on our ARShift splits.

| | | Novel Set 1 | | | | | Novel Set 2 | | | | | Novel Set 3 | | | | |
| --- | --- | --- | --- | --- | --- | --- | --- | --- | --- | --- | --- | --- | --- | --- | --- | --- |
| | Method | shot=1 | 2 | 3 | 5 | 10 | shot=1 | 2 | 3 | 5 | 10 | shot=1 | 2 | 3 | 5 | 10 |
| Ours | TFA w/ fc + CoRPN's | **40.8** | **44.8** | **45.7** | 53.1 | 54.8 | 20.4 | 29.2 | 36.3 | 36.5 | 41.5 | 29.4 | 40.4 | 44.7 | 51.7 | 49.9 |
| | TFA w/ cos + CoRPN's | **44.4** | 38.5 | 46.4 | 54.1 | 55.7 | 25.7 | 29.5 | 37.3 | 36.2 | 41.3 | 35.8 | 41.8 | 44.6 | 51.6 | 49.6 |
| | DeFRCN + CoRPN's | **44.1** | 56.8 | 62.2 | 66.0 | 65.5 | 31.8 | 41.2 | 46.1 | 49.9 | 53.3 | 38.9 | 51.3 | 54.6 | 59.4 | 61.5 |
| Main baselines | TFA w/ fc (Wang et al., 2020) | 36.8 | 29.1 | 43.6 | **55.7** | **57.0** | 18.2 | 29.0 | 33.4 | 35.5 | 39.0 | 27.7 | 33.6 | 42.5 | 48.7 | **50.2** |
| | TFA w/ cos (Wang et al., 2020) | 39.8 | 36.1 | 44.7 | **55.7** | **56.0** | 23.5 | 26.9 | 34.1 | 35.1 | 39.1 | 30.8 | 34.8 | 42.8 | 49.5 | **49.8** |
| | DeFRCN (Qiao et al., 2021) | 43.8 | **57.5** | 61.4 | 65.3 | **67.0** | 31.5 | 40.9 | 45.6 | **50.1** | 52.9 | 38.2 | 50.9 | 54.1 | 59.2 | **61.9** |
| Other baselines | FRCN+ft-full (Wang et al., 2020) | 15.2 | 20.3 | 29.0 | 40.1 | 45.5 | 13.4 | 20.6 | 28.6 | 32.4 | 38.8 | 19.6 | 20.8 | 28.7 | 42.2 | 42.1 |
| | MPSR (Wu et al., 2020) | 41.7 | 42.5 | 51.4 | 55.2 | 61.8 | 24.4 | 29.3 | 39.2 | 39.9 | 47.8 | 35.6 | 41.8 | 42.3 | 48.0 | 49.7 |
| | NP-RepMet (Yang et al., 2020) | 37.8 | 40.3 | 41.7 | 47.3 | 49.4 | **41.6** | **43.0** | 43.4 | 47.4 | 49.1 | 33.3 | 38.0 | 39.8 | 41.5 | 44.8 |
| | CME (Li et al., 2021a) | 41.5 | 47.5 | 50.4 | 58.2 | 60.9 | 27.2 | 30.2 | 41.4 | 42.5 | 46.8 | 34.3 | 39.6 | 45.1 | 48.3 | 51.5 |
| | SRR-FSD (Zhu et al., 2021) | 47.8 | 50.5 | 51.3 | 55.2 | 56.8 | 32.5 | 35.3 | 39.1 | 40.8 | 43.8 | **40.1** | 41.5 | 44.3 | 46.9 | 46.4 |

Table 12: CoRPN's not only improve performance on our ARShift split, but also perform comparably with state of the art on more traditional splits: Few-shot detection (AP50) on PASCAL VOC novel classes under three base/novel splits in the generalized few-shot learning setting. CoRPN's outperform the main baselines TFA and DeFRCN mostly in the very low shots, with comparable performance in the higher shots, regardless of classifier choice. Note that these other baselines address different aspects of few-shot detection, and could be combined with them for further improvements. All models are based on Faster R-CNN with a ResNet-101 backbone, and follow the evaluation procedure in Wang et al. (2020). Results in **bold** are the better result between ours and the main baselines.

| | | | 5-shot | | | 10-shot | | | 30-shot | | |
| --- | --- | --- | --- | --- | --- | --- | --- | --- | --- | --- | --- |
| | Method | Backbone | AP | AP50 | AP75 | AP | AP50 | AP75 | AP | AP50 | AP75 |
| Ours | TFA w/ fc + CoRPN's | ResNet-101 | **8.9** | **16.9** | **8.6** | 10.5 | **20.2** | 9.8 | 13.5 | **25.0** | 12.9 |
| | TFA w/ cos + CoRPN's | ResNet-101 | **8.8** | **16.4** | **8.7** | 10.6 | **19.9** | **10.1** | 13.9 | **25.1** | **13.9** |
| | DeFRCN + CoRPN's | ResNet-101 | **14.1** | **25.2** | **13.6** | 16.7 | **29.7** | 16.8 | 20.8 | **39.6** | 20.9 |
| Main baselines | TFA w/ fc (Wang et al., 2020) | ResNet-101 | 8.4 | 16.0 | 8.4 | 10.0 | 19.2 | 9.2 | 13.4 | 24.7 | **13.2** |
| | TFA w/ cos (Wang et al., 2020) | ResNet-101 | 8.3 | 15.3 | 8.0 | 10.0 | 19.1 | 9.3 | 13.7 | 24.9 | 13.4 |
| | DeFRCN (Qiao et al., 2021) | ResNet-101 | 13.5 | 24.7 | 13.0 | **16.7** | 29.6 | 16.7 | **21.0** | 36.7 | **21.4** |
| Other baselines | FsDetView (Xiao & Marlet, 2020) | ResNet-101 | 8.1 | 20.1 | 4.4 | 10.7 | 25.6 | 6.5 | 15.9 | 31.7 | 15.1 |
| | FSCE (Sun et al., 2021) | ResNet-101 | – | – | – | 11.9 | – | 10.5 | 16.4 | – | 16.2 |
| | CME (Sun et al., 2021) | ResNet-101 | – | – | – | 15.1 | 24.6 | 16.4 | 16.9 | 28.0 | 17.8 |
| | SRR-FSD (Zhu et al., 2021) | ResNet-101 | – | – | – | 11.3 | 23.0 | 9.8 | 14.7 | 29.2 | 13.5 |

Table 13: When there are many shots, the impact of using CoRPN's on the conventional split is reduced, likely because missing objects in the RPN stage becomes a less important effect. CoRPN's perform acceptably for few-shot detection performance on the difficult COCO novel classes task for 5-shot, 10-shot, and 30-shot. *Model re-evaluated using the standard procedure (with base and novel classes joint space) for a fair comparison. '–' denotes that numbers are not reported in the corresponding paper. CoRPN's consistently outperform the main baseline DeFRCN except for the 30-shot case.

**Bootstrapping.** Table 1 in the main paper shows that our CoRPN's outperform naive ensembles. Here we further compare CoRPN's with a bootstrapping baseline in Table 14. We construct the bootstrapping baseline by using multiple RPN classifiers, the number of which is the same as CoRPN's. Instead of selecting the most confident RPN to get gradients, we randomly select an RPN to get gradients during training. There are no additional loss terms in training these RPN classifiers. In the fine-tuning stage and the inference time, we use the same selection procedure as CoRPN's (*i.e.*, using the most certain RPN).

**Diversity loss in Dvornik et al. (2019).** In Table 14, we also compare our CoRPN's with (Dvornik et al., 2019), which utilizes a pairwise cosine similarity based diversity loss. Our CoRPN's also outperform this baseline with a considerable margin.

**Larger RPN.** We compare CoRPN's with a baseline with larger RPN sub-networks in Table 15. We find that using larger RPN sub-networks does not improve performance, suggesting that the advantage of CoRPN's is *not simply the result of using more parameters*. In CoRPN's, *all RPN classifiers share the same RPN feature extractor*, as shown in Figure 4 (main paper). We enlarge the feature dimension in the RPN. We implemented two options: a large RPN where the RPN feature extractor's output channels increase from 256 to 272, and a larger RPN where the output channels double to 512. We also modified the RPN classifier and bounding box regressor to take in larger features for both options. Table 15 shows that simply enlarging the model capacity and the anchor density of the original RPN *cannot* improve the few-shot detection performance.

| Method | AP50 |
|---|---|
| 2 RPN's, Bootstrapping | 31.8 |
| 2 RPN's, Dvornik et al. (2019) | 32.4 |
| 2 RPN's, CoRPN's (Ours) | **35.8** |

Table 14: Our diversity term – the log-determinant loss – offers improvements over a bootstrapping baseline and a pairwise cosine similarity based diversity loss in Dvornik et al. (2019). The table shows novel class AP50 under PASCAL VOC novel split 3, shot 1.

| Method | # param added | AP50 |
|---|---|---|
| TFA, original | 0 | 30.8 |
| TFA, Large RPN | 28×256 | 25.9 |
| TFA, Larger RPN | 268×256 | 27.8 |
| TFA + CoRPN's, 2 RPN's | 3×256 | **35.8** |
| TFA + CoRPN's, 5 RPN's | 12×256 | **34.8** |
| TFA + CoRPN's, 10 RPN's | 27×256 | **35.7** |

Table 15: Performance improvements of CoRPN's are not due to the increased number of parameters in CoRPN's. This table shows the novel class AP50 of TFA (Wang et al., 2020) with large RPN and even larger RPN, and compares with CoRPN's with different numbers of RPN's. All results are with PASCAL VOC novel split 3, shot 1. The second column presents how many additional parameters are introduced to the original TFA model. Using larger RPN sub-networks does not improve the performance, while CoRPN's significantly improve with fewer added parameters.

## G  ADDITIONAL QUALITATIVE RESULTS

In Figure 6 we show some additional qualitative result of our CoRPN's model on COCO. Notably, our model could successfully discover novel objects omitted by the baseline, and also successfully classify some novel objects that cause confusion for the baseline method.

## H  DATA SPLITS

**PASCAL VOC Split.** For our proposed PASCAL VOC-ARShift split, the base classes include *"bicycle", "boat", "car", "cat", "chair", "diningtable", "dog", "person", "sheep", "tvmonitor", "bird", "pottedplant", "cow", "motorbike", and "sofa"*; the novel classes include *"train", "bottle", "aeroplane", "horse", and "bus"*.

**COCO Split.** For our proposed COCO-ARShift split, the base classes include the 20 VOC classes in COCO; the novel classes include *"hot dog", "tennis racket", "fire hydrant", "laptop", "suitcase", "frisbee", "teddy bear", "bowl", "kite", and "elephant"*.

**LVIS Splits.** For our proposed LVIS-ARShift splits, the base classes include the 20 VOC classes in LVIS; the novel classes of the first split include *"Loafer (type of shoe)", "batter (food)", "cabin car", "cylinder", "egg roll", "liquor", "nailfile", "plow (farm equipment)", "vinegar", and "yoke (animal equipment)"*; the novel classes of the second split include *"broach", "burrito", "cargo ship", "crayon", "incense", "peeler (tool for fruit and vegetables)", "pin (non jewelry)", "roller skate", "tinsel", and "vodka"*; the novel classes of the third split include *"ax", "beaker", "ferry", "fish (food)", "funnel", "incense", "needle", "space shuttle", "stepladder", and "vulture"*.

