# OpenReview forum: "Improving Aspect Ratio Distribution Fairness in Detector Pretraining via Cooperating RPN’s"
_ICLR.cc/2023/Conference — Submitted to ICLR 2023_

### Official Review · Reviewer_ZdBo · 2022-10-21

**Confidence:** 4
**Correctness:** 3
**Technical Novelty And Significance:** 3
**Empirical Novelty And Significance:** 2
**Recommendation:** 3

**Clarity, Quality, Novelty And Reproducibility:**

- Reproducibility: The authors provide details about the novel splits between base and novel categories, as well as various hyper-parameters. I still think, code will be needed to fully reproduce the training strategy and the loss functions.
- Overall, the quality of the paper is good: well-written, reasonable baselines; the experimental setting can be improved
- Figures 1 and 2, and Table 1 are only referred to in the text in Section 3.2 on page 5. Figure 3 is referred to earlier. The order of figures/tables or the text should be adjusted.
- Is there a reference for the "OR" strategy? Does it stand for something specific or just literally the word "or"?
- Is "unfairness" the right word to use here? I guess bias is better, no?
- Caption of Figure 2: "different classes might have quite different bounding box distributions" ... even within one category, the aspect ratio can change significantly based on pose, consider a bicycle.

**Strength And Weaknesses:**

Strengths:
- Two different few-shot object detectors are used (DeFRCN and TFA) to showcase the benefits of the novel RPN
- The overall research direction is important I think, because it studies the practical situation of distribution shifts for novel categories in a few-shot setting.
- The results on Pascal VOC show a big gain over existing baselines.
- The analysis of the aspect ratio distributions is interesting in general, even beyond this specific few-shot learning setting.
- The ensemble baseline (and the experiment with larger RPNs in the appendix) are a fair comparison point.
- The additional experiments in the appendix are good in general, I think

Weaknesses:
- The paper's scope is limited to two-stage detectors with RPNs, and RPNs trained on the base categories for the few-shot setting. Other detector designs (like RetinaNet or DETR) are only discussed in related work, but the scope of the actual problem addressed in this paper (distribution shift of aspect ratios) should also include some form of (quantitative) experiments for these detector designs. The statement "... but likely occurs in one-stage detection too" at the end of page 2 is insufficient in my opinion.
- The experimental setting needs more justification and analysis. I understand that distribution shifts can obviously happen between base and novel categories. But I'm wondering if the aspect ratio distribution shift here is only an artifact of the small number of base and novel categories in this specific experimental setting. As far as I understood, the number of base categories is only 15, 20 and 20 for the datasets Pascal VOC, COCO, and LVIS. Why is that, when datasets COCO and LVIS have 80 and over 1200 categories, respectively? It would be good to study the distribution shift under varying number of base categories. What would happen if only five more base categories are added?
- The topic of the paper is rather narrow for a much broader problem, i.e., distribution shift between base and novel categories. The paper only addresses aspect ratio distributions for two-stage detectors. While authors comment on object location and scale, another distribution shift is due to semantic and appearance differences. Related work on open-vocabulary detection [A,B] can be discussed, where RPNs are evaluated on novel categories in a zero-shot experiment.
- The improvements for COCO and LVIS (Tables 2 & 3) are not as obvious as for the Pascal VOC dataset (Table 1). Compared to the plain ensemble, the proposed RPNs do not show a clear edge. Why is that? Is it related to the number of base and novel categories in the datasets? For VOC, there are only 5 novel categories. How do the per-category results look like?

References:
- [A] Open-Vocabulary Object Detection via Vision and Language Knowledge Distillation. Gu et al. ICLR'22
- [B] Exploiting Unlabeled Data with Vision and Language Models for Object Detection. Zhao et al. ECCV'22

**Summary Of The Paper:**

This submission is about few-shot object detection with a distribution shift of the bounding box aspect ratios between base and novel categories. The paper first elaborates that there can be splits of base and novel categories that do show a clear distribution shift of the bounding box aspect ratios and that existing few-shot object detectors perform poorly in this scenario. Then, a novel region proposal mechanism for a two-stage object detector is proposed that consists of an ensemble of foreground/background classifiers and a new training strategy. Specifically, the ensemble is trained without explicitly assigning certain aspect ratios to certain RPNs, but the RPNs "select" the boxes based on their predictive confidence. Additionally, a diversity loss and a cooperation loss are used for training. Experiments are conducted on specifically selected splits of base and novel categories for the Pascal VOC, COCO and LVIS datasets.

**Summary Of The Review:**

I'm leaning toward rejection because of the narrow problem setting, solution and experimental evaluation. Distribution shifts in few-shot learning go beyond aspect ratios, and a more thorough experiments and analyses are needed (impact of number of base categories, evaluation on larger sets of novel categories, per-category results, ...) to convince me of the importance of this problem.

---

> ### Author Response · Authors · 2022-11-25
> **Response to Reviewer ZdBo**
>
> We thank the reviewer for the valuable comments. We address all the concerns as below.
>
> **Q1**: Investigation for other detector designs (e.g., RetinaNet or DETR).
>
> **A1**: We agree with the reviewer that the problem we investigate here is broader, and likely exists in other types of detectors. Practically, most current state-of-the-art few-shot object detectors, e.g., ([1], [2], and [3]) mainly use the two-stage architecture. Therefore, to be consistent with prior work as well as for a fair comparison, we focused on improving the model performance on novel aspect ratio distributions for models equipped with RPN especially.
>
> **Q2**: Distribution shift may only be due to the small number of base categories.
>
> **A2**: We argue that there can always be a distribution shift even with more base classes. Note that, **we focus on a shift in distribution of object aspect ratios, rather than a shift of object aspect ratios**. That is, we are not specifically focusing on scenarios where novel classes have a completely new set of aspect ratios that do not appear in base classes (as in the latter case). Our shift in distribution of aspect ratios cannot be addressed by simply introducing more base classes, because such distribution shift can be caused by: (1) the existence of extreme or rare aspect ratios in some novel classes, or (2) an overall change of distribution between base and novel classes, even though they have similar types of object aspect ratios. In practice, one really should not expect novel classes to follow the same distribution of aspect ratios as base classes
>
> We are currently working on experiments under a varying number of base classes, and will include the results in the final revision. If time allows, we will provide them by the end of this discussion period.
>
> **Q3**: The topic is narrow; may consider other distribution shifts like semantic and appearance, or open-vocabulary detection [4, 5].
>
> **A3**: While we agree that there are other types of distribution shifts such as semantic and appearance differences, we kindly disagree that the topic of our paper is narrow as we do not consider other distribution shifts. In fact, for our task of few-shot detection, there is an inherent distribution shift in semantics and appearance from base to novel classes (i.e., base and novel classes are different semantic categories and thus have very different appearances). Also, as the reviewer mentioned, these other types of distribution shifts have been already studied in prior work. By contrast, we focus on the severely under-explored yet critical box aspect ratio distribution shift problem in few-shot detection, for which there has been no previous work. Simultaneously addressing multiple types of distribution shifts is an excellent direction for future research.
>
> We thank the reviewer for the related work on open-vocabulary detection [4, 5], and will cite them in the revision. Our work is orthogonal to [4, 5], as [4, 5] do not deal with box aspect ratio distribution shifts between seen and unseen classes. Also, [4, 5] explore joint vision-language models, while our study is from a pure vision perspective. Specifically, open-vocabulary detection is investigated in [4] via distilling the knowledge from vision-language models to the RCNN classifier. They directly apply a conventional RPN, which could be further improved by adopting our CoRPN’s, especially under the box aspect ratio distribution shift. An approach using vision-language models to generate pseudo-labels for open-vocabulary and semi-supervised detection is explored in [5]. They discover that repeated refinement of RPN proposals improves their quality, which is orthogonal and complementary to our CoRPN’s.
>
> **Q4**: Improvement on COCO and LVIS is not as obvious as Pascal VOC.
>
> **A4**: First of all, we would like to emphasize that while the absolute performance improvements of our CoRPN’s over state-of-the-art DeFRCN on COCO and LVIS are not as very pronounced as on Pascal VOC, they are still significant – 1.6 AP improvement for the challenge 1-shot case on COCO-ARShift with the relative performance improvements of 22.5%; 2.7/8/4.7 AP improvements for 3 splits on LVIS-ARShift with the relative performance improvements of 28.1%/100%/52.2%.

---

> > ### Author Response · Authors · 2022-11-25
> > **Response to Reviewer ZdBo, Cont'd**
> >
> > **Q5**: Per-category results on VOC.
> >
> > **A5**: We provided the per-category results as requested in Table 8 of Appendix B.
> >
> >
> > **Q6**: Clarity questions:
> >
> > **A6**:
> > Reproducibility: code was included in the original downloadable supplementary material.
> > Thanks for the comments on paper quality and settings.
> > We mention Figures 1 and 2 earlier than Figure 3 corresponding to their current order.
> > The “OR” literally means the word “or,” meaning that we do not require a consensus from all RPNs.
> > Wording: Changed “unfairness” to “bias” in the introduction.
> > Caption of Figure 2: We do agree with the reviewer’s comment. However, here we aim to emphasize that different classes may tend to have a more severe box distribution shift.
> >
> > References:
> >
> > [1] Fan et al., Few-Shot Object Detection with Model Calibration. In ECCV 2022
> >
> > [2] Pei et al., Few-Shot Object Detection by Knowledge Distillation Using Bag-of-Visual-Words Representations. In ECCV 2022
> >
> > [3] Wu et al., Multi-Faceted Distillation of Base-Novel Commonality for Few-shot Object Detection. In ECCV 2022
> >
> > [4] Gu et al., Open-Vocabulary Object Detection via Vision and Language Knowledge Distillation. In ICLR 2022
> >
> > [5] Zhao et al., Exploiting Unlabeled Data with Vision and Language Models for Object Detection. In ECCV 2022

---

> > ### Comment · Reviewer_ZdBo · 2022-11-28
> > **Re: Response to Reviewer ZdBo**
> >
> > Dear authors. I appreciate the detailed response and it answered many of my questions and concerns. Unfortunately, overall, the experimental setting still is not convincing to me for the argument the paper tries to make (overcoming distributional shift in aspect ratios). I think the problem that is addressed in this submission only exists because of an artifact of the experimental setting for few-shot object detectors (only few base and novel categories). I may be wrong, but both the authors and me make a hypothesis about this argument, and experimental evidence is needed. Hence, I think the experiment with a varying number of base categories is crucial. One data point I can share is from [A], which evaluates a zero-shot RPN on COCO and LVIS with rather small drops in performance, especially for LVIS with many base categories. Plus, the setting in [A] is zero-shot, so drops in performance may have other sources besides differences in aspect ratio.
> >
> > [A] Open-vocabulary Object Detection via Vision and Language Knowledge Distillation. Gu et al. ICLR 2022

---

> > > ### Author Response · Authors · 2022-12-13
> > > **Additional Experiment with More Base Classes**
> > >
> > > We thank reviewer ZdBo for the question. We present the additional experiment with more training classes requested by the reviewer: In ARShift-COCO, we keep the 10 test classes unchanged, and expand the training classes from 20 to the rest 70 classes. We argue the problem of aspect distribution shift persists in this setting with more training classes. Despite a performance improvement, the vanilla DeFRCN still underperforms our CoRPN’s on 1-shot and 2-shot. Therefore, the proposed problem of box aspect ratio distribution shift is not due to a relatively small number of training classes in our setting.
> > >
> > > |                         | 1-shot      | 2-shot       |
> > > |-------------------------|-------------|--------------|
> > > | DeFRCN + CoRPN’s (Ours) | **8.7 -> 10.4** | **12.4 -> 14.2** |
> > > | DeFRCN                  | 7.1 -> 9.7  | 11.5 -> 13.8 |

---

### Official Review · Reviewer_UxEa · 2022-10-23

**Confidence:** 5
**Clarity, Quality, Novelty And Reproducibility:** The clarity, quality, novelty and rep…
**Correctness:** 3
**Technical Novelty And Significance:** 3
**Empirical Novelty And Significance:** 3
**Recommendation:** 5

**Strength And Weaknesses:**

Strength
1. The motivation is interesting, which aims to solve the aspect ratio gap problem.
2. New ARShift benchmarks are useful for future research on aspect ratio.
3. The proposed CoRPN is simple yet effective.

Weaknesses
1. Is CoRPN only designed for few-shot object detection? If so, it is better to add 'few-shot' to the title. If not, can you provide more results on other detection settings?
2. Can your show the aspect ratio distributions of RPN and CoRPN?  It is important to prove that your CoRPN generates more diverse anchors than RPN.
3. ARShift benchmarks artificially enlarge aspect ratio gaps between the training and test sets, which is somewhat tricky. Can you show the results on standard detection benchmarks?

**Summary Of The Paper:**

This paper aims to solve the aspect ratio gap between base and novel classes in few-shot object detection. The authors present a very simple CoRPN method, which only uses multiple classifiers in RPN to generate more diverse anchors. To evaluate the effects of the aspect ratio gap, the authors propose ARShift benchmarks, which enlarge the aspect ratio differences between the training and testing sets in standard detection benchmarks. Experimental results show that the proposed CoRPN achieves remarkable improvements on ARShift benchmarks.

**Summary Of The Review:**

Overall, it's a good work, simple yet effective. My main concern is experiments. The authors only test on self-made benchmarks.

---

> ### Author Response · Authors · 2022-11-25
> **Response to Reviewer UxEa**
>
> We thank the reviewer for the valuable comments. We address all the concerns as below.
>
> **Q1**: Whether CoRPN is designed for few-shot detection. Results on general detection settings.
>
> **A1**: Yes, our work focuses on few-shot detection. We have added “few-shot” to the title in the revision. The issue of box aspect ratio distribution shift is especially severe in few-shot detection. This is because 1) in few-shot detection, one really should not expect novel classes to follow the same distribution as base classes; 2) unlike general detection, few-shot detection requires improved generalization from the base detector as there are very few examples of the novel classes. Therefore, our CoRPN’s are proposed to primarily address few-shot detection.
>
> On the other hand, while for general detection with large amounts of annotated data there are no novel classes and thus no shifted distribution, our CoRPN’s still perform comparably. We tested integrating our CoRPN’s to Detectron2’s baseline Faster-RCNN model for object detection on the conventional COCO benchmark. Detection AP is almost the same: w/o: 37.9 -> with CoRPN’s: 38.0 mAP.
>
> **Q2**: Aspect ratio distribution of CoRPN’s and RPN
>
> **A2**: Please refer to Figure 5 in the Appendix. Our CoRPN’s generate a **multi-modal** aspect ratio distribution, while the conventional RPN generates a **unimodal** distribution. This shows that our CoRPN’s improve the aspect ratio diversity of conventional RPN and hence become more robust to aspect ratio distribution shift.
>
> **Q3**: Results on standard detection benchmarks
>
> **A3**: First, CoRPN’s perform comparably with or better than state of the art on traditional splits of few-shot detection (proposed in the prior work). We evaluated this in Table 4 of the original manuscript. We observe that CoRPN’s mostly beat strong baselines in few-shot detection performance on the (extremely difficult) COCO novel classes task for 1, 2, and 3-shot cases. We also provided more results on traditional splits in Section E of the original Appendix.
>
> Second, as mentioned in Q1/A1, CoRPN’s still perform comparably in general detection with large amounts of annotated data.
>
> These results validate that CoRPN’s are consistently effective in different detection settings, with significantly larger performance gains when the distribution shift becomes more severe. Because of such generalizability of CoRPN’s, it is good to always use our approach, no matter which splits.

---

> > ### Comment · Reviewer_UxEa · 2022-11-28
> > **Reviews**
> >
> > Thanks for the authors' responses.
> >
> > I have checked the comments from other reviewers and the corresponding responses.
> > I agree with other reviewers. Although the motivation of this paper is interesting (the authors show the importance of aspect ratios in object detection), the current version is only effective for a very limited scenario (RPN in two-stage architectures for few-shot detection). It would be much better to generalize the method for more detection scenarios.

---

### Official Review · Reviewer_NdSa · 2022-10-24

**Confidence:** 5
**Correctness:** 2
**Technical Novelty And Significance:** 3
**Empirical Novelty And Significance:** 2
**Recommendation:** 3

**Clarity, Quality, Novelty And Reproducibility:**

I cannot see the attached code so I cannot verify the reproducibility of the paper. The paper seems not easy to implement.

**Strength And Weaknesses:**

*Strengths*
+ The finding is quite interesting, i.e., shifting in aspect ratio distribution of object bounding boxes between the base and novel classes.
+ The structure of this paper is well-written.

*Weaknesses*
+ Incorrect average results in Tab. 3. For DeFRCN, the AP for split 1, 2, and 3 are 9.6, 8.0, and 9.0, respectively. But they report the overall AP is 15.0. The same observation applies to the DeFRCN Ensemble of Experts. Thus, the conclusions drawn are wrong for this table. DeFRCN Ensemble of Experts does help a lot.
+ This method aims to address the poor proposal generation problem of RPN. It would be much better to include the AR results to prove their recall is better than that of the baseline.
+ The authors should explain the intuition behind the formulation of diversity loss. It is not trivial to interpret the proposed loss to enhance the diversity of the classifiers.
+ Do the authors claim the new split of COCO and LVIS datasets as a contribution? The improvement in the original splits seems very marginal. If the base classes contain high-variation in the bounding box aspect ratios, e.g. human, they can arguably cover the aspect ratio of many novel classes.
+ Since the paper mainly focuses on the improvement of RPN, they should move the qualitative results to the main paper, not the supplementary material.
+ Fig. 1 does not reflect the motivation of this paper. The proposals on the right are still good which can be further regressed in the second stage of Faster-RCNN.


**Summary Of The Paper:**

This paper addresses the problem of the aspect-ratio shift between the base and novel classes in few-shot object detection resulting in the poor box proposals of the RPN in a two-stage detector. They propose using multiple classifiers instead of one as in the current RPN. Their experiments show significant improvements with the new versions of the COCO and LVIS datasets but marginal improvements with the original datasets. The new splits of the COCO and LVIS datasets are created by re-select the subset of base and novel classes whose aspect ratios are significantly different.

**Summary Of The Review:**

The paper proposed a method for alleviating the problems of aspect ratio shift in the few-shot object detection by introducing multiple classifiers of the RPN instead of one. It is arguable to see the practicality of the new splits of the COCO and LVIS datasets since the proposed methods have marginal improvements in the common setting. Also, the paper also has the incorrect average results reported thus drawing an incorrect observation of the ensemble RPN.

---

> ### Author Response · Authors · 2022-11-25
> **Response to Reviewer NdSa**
>
> We thank the reviewer for the valuable comments. We address all the concerns as below.
>
> **Q1**: Average results in Tab. 3
>
> **A1**: Our results in Tab. 3 were correct. Note that the last column is “Entire test set” rather than “average” – we apologize for the confusion. Specifically, for the LVIS result shown in Tab. 3, “splits 1, 2, and 3” show the performance for our three curated test sets, each including 10 classes. As mentioned in the caption, the column “Entire test set” shows the average performance for all rare categories in the LVIS test classes. Therefore, the combination of our three splits does not cover the “Entire test set,” so averaging the performance for the three splits will not result in the numbers reported in the last column.
>
> **Q2**: AR results.
>
> **A2**: Following the reviewer’s suggestion, we present the AR1000 results on the three splits in our LVIS-ARShift benchmark, as shown in the table below and also in the revision. Our CoRPN’s outperform both baselines in recall.
>
> |                            | Split 1 | Split 2 | Split 3 |
> |----------------------------|---------|---------|---------|
> | DeFRCN + CoRPN's (Ours)    | 18.4    | 24.8    | 28.3    |
> | DeFRCN                     | 17.6    | 18.5    | 13.9    |
> | DeFRCN Ensemble of Experts | 18.1    | 23.4    | 25.0    |
>
>
> **Q3**: Intuition behind diversity loss.
>
> **A3**: A log-determinant loss forces a set of vectors to have full rank, and is widely used as a barrier in interior point methods for semidefinite programs (e.g., [1]); there is a long tradition of using log-determinant divergences in machine learning (e.g., [2]). In our case, we construct an N by NA matrix of probabilities, whose element represents the foreground probability produced by one of the N RPNs on one of the NA anchor boxes. Intuitively, the log-determinant loss encourages the probability matrix to have rank N, so each RPN is reacting differently on the collection of NA boxes, thus enhancing the diversity of the N RPNs.
>
> **Q4**: Whether the new splits are a contribution. Base class covers diverse aspect ratios.
>
> **A4**: First, we claim the new splits, i.e., the ARShift benchmarks of PASCAL VOC, COCO, and LVIS datasets, as one of our contributions. As acknowledged by Reviewer UxEa, “ARShift benchmarks are useful for future research.” Our work reveals a problem that is grounded in real-world applications yet substantially overlooked in the current few-shot learning literature: one really should not expect novel classes to follow the same distribution of aspect ratios as base classes. To quantitatively study this problem and provide an effective solution, we propose the ARShift benchmarks with the new, hard splits.
>
> Second, while the improvements of our CoRPN’s in the conventional splits are marginal, our CoRPN’s **consistently achieve the best performance in both hard and random conventional splits**, with more pronounced improvements on the more challenging hard splits. We would like to further emphasize the practical importance of evaluation on hard splits. In practice, we cannot assume that the split an application offers is necessarily easy, or, for that matter, uniform and random. Notice the established tradition of worst-case reasoning in safety-critical applications. An autonomous vehicle or a medical system should likely be evaluated on its performance on challenging splits rather than on the average split.
>
> Third, we would like to clarify that our problem cannot be addressed by including base classes of high-variation in the bounding box aspect ratios. This is because we focus on a shift in distribution of object aspect ratios between base and novel classes, rather than merely a shift of object aspect ratios. That is, we are not specifically focusing on scenarios where novel classes have a completely new set of aspect ratios that do not appear in base classes (as in the latter case). Instead, a shift in distribution of aspect ratios – what we study – can be caused by: (1) the existence of extreme or rare aspect ratios in some novel classes, or (2) an overall change of distribution between base and novel classes, even though they have similar types of object aspect ratios. Therefore, covering diverse aspect ratios in base classes (e.g., human) does not solve the problem. In fact, the human class mentioned by the reviewer is always included in our base classes in all settings and the problem persists.
>
> **Q5**: Move qualitative results to the main paper.
>
> **A5**: Thanks for the suggestion. We have moved the first four results to Figure 4 in the main paper.

---

> > ### Author Response · Authors · 2022-11-25
> > **Response to Reviewer NdSa, Cont'd**
> >
> > **Q6**: Fig. 1 does not reflect the motivation. In Fig. 1 the proposals on the right are still good.
> >
> > **A6**: We apologize for the confusion. **The proposals on the right are generated by our CoRPN’s’** and their quality is good. The image on the left shows the unmodified RPNs in DeFRCN could not generate any proposal with IoU > 0.7 with the ground truth box. So Fig. 1 shows the box aspect ratio distribution shift problem we try to address, which reflects our motivation. We modified the caption of the figure to emphasize that more in the revision.
> >
> > **Q7**: Clarity question: code
> >
> > **A7**: Code was attached in the original downloadable supplementary material.
> >
> > References:
> >
> > [1] Boyd, S., Vandenberghe, L.: Convex optimization (2009)
> >
> > [2] Dhillon, I.S.: The log-determinant divergence and its applications (2008)

---

### Official Review · Reviewer_haU4 · 2022-10-24

**Confidence:** 5
**Correctness:** 3
**Technical Novelty And Significance:** 1
**Empirical Novelty And Significance:** 1
**Recommendation:** 1

**Clarity, Quality, Novelty And Reproducibility:**

- As mentioned in the weaknesses, the manuscript must be revised thoroughly to improve its clarity and presentation quality.
- As for novelty, it doesn't seem to meet the ICLR acceptance criteria.
- Since the important implementation details (i.e., the number of RPNs) is missing, it is difficult to be reproduced.

**Strength And Weaknesses:**

* Strengths
1. For the test environment using the ARShift benchmark, it was effective to significantly increase the few-shot detection accuracy using the proposed method on the three detection datasets. It also prevented performance degradation for the base categories, which is often caused by the catastrophic forgetting problem that is common in a few-shot detections.


* Weaknesses
1. The problem in a specific environment of the few-shot detection problem, which has a large difference in the aspect ratio distribution of the basic category and the novel category, as claimed in this paper, is of interest to only a few researchers, and the applicability/generality of the problem is very Limited. Thus, the environment claimed in this paper did not appear in the general dataset, but only in the dataset that was modified to emphasize the environment.

2. In addition, the proposed method using multiple RPNs is not designed to address a claimed problem. Experimentally, several RPNs were specialized in generating region proposals of specific aspect ratios (Tab 5), but this function was not considered at all in the CoRPN design.

3. Paper presentation is immature yet.
- Figure 1 is not mentioned anywhere in the manuscript's content.
- The same explanation about using Faster R-CNN are given in the last paragraph of Section 3.1 and the second paragraph of Section 3.3.
- In Figure 2, it's hard to figure out what 'density' is as the sum of the densities over the entire aspect ratio range seems to be greater than 1.
- In the top row of Figure 3, there are no explanations about the difference betwee the gray and blue box. In addition, non-maximum suppression is applied after performing bouding box regression so the illustrations in the bottom row of Figure 3 may bring incorrect information. Most of all, this figure does not deliver important information and nor is hard to understand as it has to occupy the large space of the manuscript.
- In eq 3, Sigma may be Sigma_jk
- Important details of experimental setup is missing such as the number of RPNs. According to Tab. 5, the number of RPNs may be larger than five.

4. Some questionable impact of the proposed method based on the experimental results
- In Tab. 5, only two or three RPNs appear to be activated. Does it mean the proposed method uses multiple RPNs inefficiently.
- What is the version of CoRPN's with the best accuracy? Is it the model trained with the Cooperation loss with phi=0.5 and the diversity loss? If so, the CoRPN seems to be very sensitive to the phi.

5. Similar problems and similar method designs have already been claimed in [a]. [a] is applied to general object detection and more experts than RPN are used for the classifier, but since it seems very simple to apply it to RPN, comparison with [a] will be necessary.

[a] H. Lee, Multi-expert R-CNN for Object Detection, In IEEE TIP.

**Summary Of The Paper:**

This paper claims that in few-shot detection tasks, performance can be degraded due to the lack of RPN ability to find object proposals for new categories with aspect ratios significantly different from those of the base categories. To overcome the limitations of RPN in few-shot detection, this paper uses multiple RPNs that have been trained in the direction of enhancing diversity among RPNs and cooperating with each other. To adequately validate this benefit of the proposed method, base/novel categories split of datasets are modified so that the (few-shot) images of the novel categories have very different aspect ratios from images of the base categories, which are referred ARShift. Object detectors trained using the proposed method successfully increases the accuracy for the novel categories, avoiding the performance degradation caused by the catastrophic forgetting issue for the base categories in the few-shot detection task on the three datasets, PASCAL VOC, COCO, and LVIS datasets.

**Summary Of The Review:**

My rating was based on the points presented in Weaknesses. My biggest concern is that there is no consistency between the problem to be claimed and the method design to solve the problem.

---

> ### Author Response · Authors · 2022-11-25
> **Response to Reviewer haU4**
>
> We thank the reviewer for the valuable comments. We address all the concerns as below.
>
> **Q1**: The problem is specific and of interest to only a few researchers.
>
> **A1**:  We respectfully disagree with the reviewer that our problem is in a specific environment. We argue that our work investigates a scenario that is **grounded in real-world applications yet substantially overlooked in the current few-shot learning literature**: one really should not expect novel classes to follow the same distribution of aspect ratios as base classes. Note that, we focus on a shift in distribution of object aspect ratios, rather than a shift of object aspect ratios. That is, we are not specifically focusing on scenarios where novel classes have a completely new set of aspect ratios that do not appear in base classes (as in the latter case). Instead, our shift in distribution of aspect ratios happens often in practice and can be caused by: (1) the existence of extreme or rare aspect ratios in some novel classes, or (2) an overall change of distribution between base and novel classes, even though they have similar types of object aspect ratios. Empirical evidence shows our scenario occurs. Figure 2 in the manuscript shows **distribution shift is universal**: on COCO and LVIS, it exists for not only our generated ARShift hard splits but also traditional splits – again, it is just that this phenomenon has been overlooked. Figure 1 in the manuscript shows that rare aspect ratios occur in this data, too.
>
> Broadly speaking, our investigation is consistent with the established tradition of **worst-case reasoning in safety-critical applications**. An autonomous vehicle or a medical system should likely be evaluated on its performance on challenging splits rather than on the average split. Failure to detect novel classes under the hard splits will cause severe issues in practice.
>
> We believe that our problem is of wide interest. Indeed, as acknowledged by Reviewer NdSa, “the finding is quite interesting, i.e., shifting in aspect ratio distribution of object bounding boxes between the base and novel classes;" by Reviewer UxEa, “ARShift benchmarks are useful for future research;” and by Reviewer ZdBo, “the overall research direction is important because it studies the practical situation of distribution shifts for novel categories in a few-shot setting.”
>
> In addition to formulating this under-explored problem and introducing the associated ARShift benchmarks, our key contributions also include proposing a modified RPN subsystem, CoRPN’s, for a standard detection pipeline which results in improved few-shot detection. **Our approach is general, and it consistently achieves the best performance on both our ARShift hard splits and conventional splits**. As mentioned by Reviewer NdSa, our CoRPN’s achieve significant improvements over the state of the art on the ARShift benchmarks, e.g., up to 7.3AP on VOC-ARShift and up to 1.6 AP on more challenging COCO-ARShift, as well as marginal improvements on the conventional setting. Because of the generalizability of CoRPN’s, it is good to always use our approach, no matter which splits.
>
> **Q2**: Proposed method is not designed for the problem. The specialization of RPNs is not considered in the design of CoRPNs.
>
> **A2**: We believe that our method aligns well with addressing our problem, because our method is able to handle a wider range of proposal regions and so handle distribution shifts between base and novel classes. Concretely, we would like to clarify from the following two aspects:
>
> (1) The key to improving the generalization of few-shot detectors under box aspect ratio distribution shifts is to make the region proposal network (RPN) *not biased* towards familiar aspect ratio distributions of base classes. This is difficult for existing few-shot detectors, as they use a single RPN which is not able to identify every likely positive box. By contrast, our CoRPN’s reduce this bias in the RPN by training multiple distinct but cooperating specialized RPNs and thus increasing/enriching the family of proposal regions the RPN subsystem can detect. Hence, this means that the detector can generalize to the shifted aspect ratio distribution of novel classes well.
>
> (2) We respectfully disagree with the reviewer that the function of several RPNs being specialized in generating region proposals of specific aspect ratios is not considered in the CoRPN’s design. While our CoRPN’s do not explicitly enforce each RPN classifier to specialize in specific aspect ratios, **such specialization emerges in the learning process**.

---

> > ### Author Response · Authors · 2022-11-25
> > **Response to Reviewer haU4, Cont'd**
> >
> > As mentioned by **Reviewer ZdBo**, “the ensemble is trained without explicitly assigning certain aspect ratios to certain RPNs, but the RPNs “select” the boxes based on their predictive confidence." Indeed, when we let the CoRPNs select boxes based on their predictive confidence, it would be easier for the CoRPNs to specialize in aspect ratio since it is more distinguishable for the model.
> >
> > More specifically, as explained in Sec. 3.3, we train our RPN classifiers using an OR strategy – a box is classified with the label reported by the most confident RPN, which gets the gradient during training. This has two effects. First, the RPN’s can specialize to a degree. In turn, if one RPN misses a positive box, the other might find it. Second, the training strategy may improve the variation of proposals, which is especially essential when dealing with a different proposal box distribution in the few-shot case. Both effects may bring improvements to the model’s robustness with respect to aspect ratio distribution.
> >
> > We see specialization emerging in the learning process as an advantage of our CoRPN’s design over hand-craft strategies.
> >
> > **Q3**: Presentation problems.
> >
> > **A3**:
> > (1) We mentioned Figure 1 in the revision.
> >
> > (2) We have reduced redundant descriptions of the RPN structure in Section 3.3.
> >
> > (3) The “density” here shows the kernel density estimate, so the bin width multiplied by this density would sum to 1 for each curve.
> >
> > (4) We have taken the reviewer’s suggestion on Figure 3 and removed the top row explaining the two-phase training for few-shot detection. For the reviewer’s information, in the top row, the different color shows that these two types of network modules are treated differently in our two baselines. The modules in green are trained normally, while the modules in gray are frozen in TFA and trained with a scaled gradient or stop gradient in DeFRCN. In the bottom row, we have added the missing arrow from box regression to NMS. We keep the bottom figures, because they show the difference between conventional RPN and our CoRPN’s.
> >
> > (5) We have changed “Sigma” to  “Sigma_{jk}”.
> >
> > (6) We used 5 RPNs and listed our hyperparameter settings in the original Appendix Section C. We believe Table 5 is not in contradiction to our hyperparameter setting – note that in Table 5, the RPNs are 1-based indexed ranging from RPN1 to RPN5.
> >
> > **Q4**: Experimental Questions.
> >
> > **A4**:
> > (1) It is not the case that only two or three RPNs are activated. Table 5 shows the RPN (or combinations of two RPNs) that achieves > 98% coverage on this set of boxes. This does not mean that other RPNs’ outputs have no coverage when they are not mentioned, despite their lower coverage.
> >
> > (2) The best hyperparameters we used were explained in the original Appendix Section C: we use phi=0.3. We used the COCO hyperparameters on LVIS directly and obtained the state-of-the-art result, showing that **our model is not sensitive to phi across datasets** and phi=0.3 is generally a good choice.
> >
> > In addition, notice that the ablation in Table 6 covers a wide range of phi values: phi should be within the range of 0 to 1 and we ablated phi from 0.1 to 0.9. Notably, when phi changes from 0.1 to 0.9, while the results of our CoRPN’s vary, **all of them outperform the baseline** – we would like to argue that this in fact shows the robustness and effectiveness of our CoRPN’s.
> >
> > Also, as mentioned in Section 3.5, phi is acting as a lower bound for each RPN’s probability assigning to a foreground box; so phi should be usually less than 0.5.
> >
> > **Q5**: Comparison with ME R-CNN [a].
> >
> > **A5**: Table VII in ME R-CNN [a] shows that ME R-CNN only marginally improves upon a “hard-coded assignment baseline” by 0.8 AP on Pascal VOC, meaning that they perform very comparably. This “hard-coded assignment baseline” is very similar to our “DeFRCN Ensemble of Experts” baseline. We improve upon this baseline in our paper by large margins of up to 7.3AP on our VOC-ARShift. Therefore, we have a good reason to believe that ME R-CNN cannot bring large improvements in our setting and significantly under-performs our method.
> >
> > Such performance discrepancy is because we have a significantly different objective than ME R-CNN. Our paper focuses on addressing the box aspect ratio distribution shift in few-shot detection, while ME R-CNN focuses on conventional detection and mixture-of-experts. Since **[a] is not particularly relevant and does not provide code**, we leave additional comparison to future work.
> >
> > [a] H. Lee, Multi-expert R-CNN for Object Detection, In IEEE TIP.
> >
> > **Q6**: Reproducibility.
> >
> > **A6**: We provided implementation details and hyperparameters in the original Appendix Section C and provided our code in the original downloadable supplementary material.

---

### Decision · Program_Chairs · 2023-01-20

**Decision:**

Reject

**Justification For Why Not Higher Score:**

Immature paper

**Justification For Why Not Lower Score:**

N/A

**Metareview: Summary, Strengths And Weaknesses:**

This paper claims that in few-shot detection tasks, performance can be degraded due to the lack of RPN ability to find object proposals for new categories with aspect ratios significantly different from those of the base categories. To overcome the limitations of RPN in few-shot detection, this paper uses multiple RPNs that have been trained in the direction of enhancing diversity among RPNs and cooperating with each other. To adequately validate this benefit of the proposed method, base/novel categories split of datasets are modified so that the (few-shot) images of the novel categories have very different aspect ratios from images of the base categories, which are referred ARShift. Object detectors trained using the proposed method successfully increases the accuracy for the novel categories, avoiding the performance degradation caused by the catastrophic forgetting issue for the base categories in the few-shot detection task on the three datasets, PASCAL VOC, COCO, and LVIS datasets.

The proposed method improves the few-shot accuracy in three detection datasets.

One of the main problems is the difference in aspect ratios. The dataset is not realistic. The paper writing is immature. Missing motivation. The paper's scope is limited to two-stage detectors with RPNs. The experimental setting needs more justification and analysis. The topic of the paper is rather narrow for a much broader problem. The improvements for COCO and LVIS (Tables 2 & 3) are not as obvious as for the Pascal VOC dataset (Table 1).

The authors addressed some of the comments. However, didn't manage to convince the reviewers. And the paper is not mature enough to be published.